# Align-VL: Can Being Modest Help in the Alignment of Vision-Language Models?

## Abstract

Multimodal alignment aims to learn a shared latent space between different modal inputs to establish connections across modalities. A prime example is Visual Language Models (VLMs), such as CLIP, which benefit from extensive image-text pre-training and excel in image recognition tasks. These models are emblematic of successful multimodal alignment. Subsequent work has successfully aligned multimodal data on limited datasets using feature mixing enhancement methods. However, these models encounter significant challenges: The presence of *ambiguous samples (either partially matched or completely unmatched)* in datasets with weakly associated, low-quality image-text pairs causes models to become overconfident (in training) and confused (in inference), ultimately reducing performance. Current contrastive learning methods, which rely on single positive pairs, exacerbate this issue by encouraging overconfidence when the model encounters such ambiguous samples. To overcome these challenges, we developed Align-VL, a multimodal alignment enhancement method that operates on the latent spaces of pre-trained unimodal encoders. This approach adjusts the matching degree of the data and moderates model overconfidence, promoting more appropriate and effective alignments. Align-VL incorporates *Random Perturbation* and *Embedding Smoothing* strategies to enhance input feature robustness and reduce model overconfidence, improving the model's ability to manage uncertainty and generalize to new data. In our experiments, Align-VL outperformed existing state-of-the-art (SoTA) methods in image-text retrieval tasks, demonstrating its superior effectiveness. Align-VL also offers significant reductions in training time and data requirements compared to methods like CLIP, using substantially fewer GPU days and image-text pairs. Code will be publicly available.

## 1 Introduciton

Multimodal learning, by integrating different types of data modalities, enhances a model's perception and understanding capabilities, facilitating cross-modal information interaction and integration Radford et al. (2021); Vouitsis et al. (2024); Zhai et al. (2022); Liu et al. (2023b;a); Yang et al. (2021); Girdhar et al. (2023). Recent advancements in multimodal machine learning have shown unprecedented potential across various application fields, with some applications even attracting mainstream attention Girdhar et al. (2023); Radford et al. (2021). The cornerstone of multimodal learning is multimodal alignment, which maps information from multiple modalities, such as text and images, into a unified multimodal vector space Radford et al. (2021); Alayrac et al. (2022). Researchers have made numerous efforts in multimodal alignment, with Visual Language Models (VLMs) being particularly representative. VLMs like CLIP Radford et al. (2021), which undergo extensive image-text pre-training, excel in image recognition tasks, showcasing the potential of VLMs in establishing effective cross-modal connections.

The success of multimodal alignment largely relies on large-scale training mechanisms like CLIP, which often require extensive GPU resources and rely on billions of multimodal data pairs Zhai et al. (2022); Radford et al. (2021); Alayrac et al. (2022). However, the high computational costs are impractical for scenarios with limited computing resources or scarce multimodal data. Therefore, designing a cost-effective and efficient multimodal alignment framework is crucial. Inspired by Mixup Zhang et al. (2018), Fusemix introduces an efficient strategy for multimodal alignment Vouitsis et al. (2024) by augmenting the latent spaces of pre-trained unimodal encoders, allowing

for model creation with significantly reduced data and computational requirements. However, ambiguous samples—whether partially matched or completely unmatched—in datasets with weakly associated (see Figure 4), low-quality image-text pairs can lead to overconfidence and confusion in models, ultimately degrading performance. Moreover, current contrastive learning methods, which rely on single positive examples, exacerbate this issue by further encouraging overconfidence in the presence of ambiguous samples Radford et al. (2021); Vouitsis et al. (2024).

To overcome these issues, building on existing foundation Vouitsis et al. (2024), we propose Align-VL, a multimodal alignment enhancement method designed to adjust the matching degree of the data and moderate model overconfidence, incorporating two key components: 1) *Random Perturbation*: This introduces normally distributed perturbations at the visual-text feature level to simulate uncertainty, enhancing the model's generalization capability and helping it learn more robust feature representations. 2) *Embedding Smoothing*: This aims to smooth the model's prediction of output distributions, moderating model overconfidence in positive samples and increasing the smoothness for predictions on uncertain samples, thereby enhancing generalization. By using Align-VL to align the latent spaces of pre-trained unimodal encoders, we have developed a highly competitive visual-language (V-L) model. In retrieval tasks, this model not only surpasses existing state-of-the-art (SoTA) methods but also significantly reduces the need for computational resources and data, as detailed in Figure 1. Our study makes two significant contributions:

- Statistical analysis reveals that the quality of existing image-text paired datasets is suboptimal, causing VLMs to become confused and overly confident when faced with ambiguous positive pairs (either partially matched (Figure 4 (a), (b), (c)) or completely unmatched (Figure 4) (d)). This significantly impacts the performance of multimodal alignment.

- We propose a novel V-L alignment algorithm, Align-VL, which incorporates *Random Perturbation* to simulate input uncertainty and *Embedding Smoothing* to mitigate overconfidence in positive samples. This Align-VL enhances model generalization and robustness, effectively addressing the challenges posed by the suboptimal quality of existing datasets.

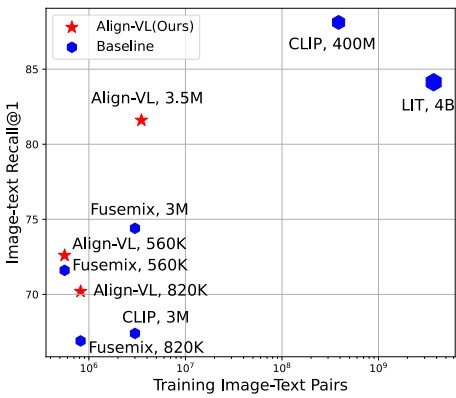

Figure 1: Image-to-Text retrieval performance on the Flickr30K test set Young et al. (2014) is plotted against the number of training pairs on a log-scale x-axis, illustrating how training volume impacts effectiveness.

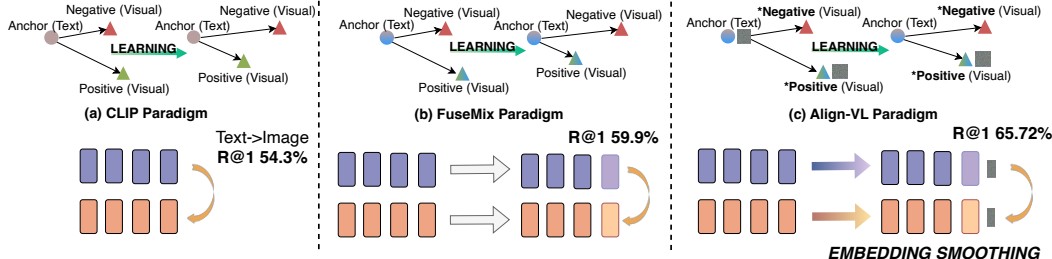

Figure 2: Three Multimodal Alignment Paradigms: CLIP for Contrastive Learning, FuseMix for Enhanced Mixing Embeddings, and Align-VL for Moderating Model Overconfidence and Enhancing Robustness. In Fusemix and Align-VL, the text anchor and visual positive sample are derived from mixed features. In Align-VL, the visual and text positive pairs are the embedding augmented with random perturbation.

## 2 RELATED WORK

Multimodal alignment achieves cross-modal synchronization not through direct correspondences between modalities, but implicitly via internal model mechanisms that discern latent semantic connections within the data. The primary objective of these models is to learn a shared latent space capable of jointly encoding multiple modalities, thereby facilitating effective multimodal alignment Tan & Bansal (2019); Li et al. (2020); Yuan et al. (2021); Wang et al. (2022a); Bao et al. (2022); Wang et al. (2022b); Girdhar et al. (2022); Likhosherstov et al. (2023); Zhang et al. (2023); Wu et al. (2023). Image-language alignment is a pivotal area of study in multimodal alignment, aiming to create universal models capable of interpreting both image and language data. Standard multimodal models usually undergo end-to-end training on image-text pairs. Yet, training these large-scale models from scratch demands substantial computational and data resources, which can restrict scalability. Arandjelovic & Zisserman (2017); Lu et al. (2019); Sun et al. (2019); Su et al. (2020); Chen et al. (2020); Li et al. (2021; 2022).

Pioneered by CLIP and ALIGN Radford et al. (2021); Jia et al. (2021), this approach uses a dual-encoder architecture, jointly embedding text and images into the same latent space through contrastive target training. 3T aligns text and image encoders with the latent space of a pretrained classifier Kossen et al. (2023). LiT uses a frozen pretrained image classifier as the image encoder and aligns a text encoder to it Zhai et al. (2022). Although these methods have seen success, they mostly train one or two encoders from scratch, relying on expensive cross-GPU gradient computations. ImageBind Girdhar et al. (2023) uses images as anchors to learn a shared latent space across six modalities through contrastive learning, jointly training various modality encoders from scratch. Moreover, the large-scale image-text paired datasets they use, ranging from 400 million to 5 billion pairs, mostly sourced from the internet, are generally not public Vouitsis et al. (2024). In contrast to these works, Fusemix boosts computational and data efficiency through feature augmentation techniques, using frozen pre-trained unimodal encoders and fewer multimodal paired data, requiring fewer resources Vouitsis et al. (2024). However, ambiguous positive samples in weakly associated datasets (see Figure 4) lead to model overconfidence and degraded performance, exacerbated by contrastive learning methods that focus on single positive examples Radford et al. (2021).

Figure 2 compares three multimodal alignment paradigms: the CLIP Paradigm, which utilizes contrastive learning to manage data point relationships; the Fusemix Paradigm, which enhances embeddings by mixing features of image-text pairs; and the Align-VL Paradigm, which reduces overconfidence through perturbations and embedding smoothing, enhancing generalization and robustness (CLIP and FuseMix train on 3M data pairs, Align-VL uses about 3.5M pairs to achieve its results). The Align-VL specifically reduces model overconfidence and ensures experiments are computationally and data efficient, requiring only a reasonable amount of GPU resources (see Figure 6).

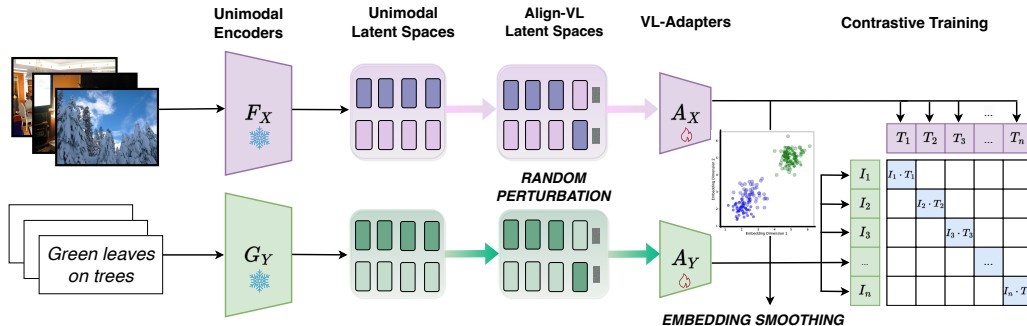

Figure 3: A pipeline of the Align-VL showcases the process of aligning the latent spaces of pre-trained unimodal encoders using a fewer dataset of paired data. The unimodal encoders remain frozen throughout the process, with their latent encodings pre-computed only once for efficiency. In this framework, both Random Perturbation and Embedding Smoothing are applied to each latent space to enhance robustness and reduce model overconfidence. Lightweight V-L adapters are then trained to meticulously align these augmented latents into a cohesive, shared latent space, effectively bridging the semantic gap between different modalities.

## 3 METHODOLOGY

In this section, we introduce the Align-VL framework, designed to facilitate visual-text modal alignment in the latent space while addressing key considerations such as model overconfidence, and computational and data efficiency. Align-VL entire process is illustrated in Figure 3.

### 3.1 PRELIMINARIES

**Notation:** We define the task of V-L alignment from an alignment perspective. The goal is to learn a shared latent space between visual and textual modal inputs. Formally, given any two data modalities (images $\mathcal{X}$ and text $\mathcal{Y}$), our objective is to learn two networks, $f_X : \mathcal{X} \to \mathcal{S}$ and $f_Y : \mathcal{Y} \to \mathcal{S}$, that embed each modality into a shared latent space $\mathcal{S}$.

We take our two encoders as $f_X = F_X \circ A_X$ and $f_Y = G_Y \circ A_Y$. That is, we define $F_X(Frozen) \colon \mathcal{X} \to \mathcal{U}_\mathcal{X}$ and $G_Y(Frozen) \colon \mathcal{Y} \to \mathcal{U}_\mathcal{Y}$, where $\mathcal{U}_\mathcal{X}$ and $\mathcal{U}_\mathcal{Y}$ are intermediate latent spaces. We then have $A_X(Learnable) \colon \mathcal{U}_\mathcal{X} \to \mathcal{S}$ and $A_Y(Learnable) \colon \mathcal{U}_\mathcal{Y} \to \mathcal{S}$, which we hereafter refer to as V-L adapters. Our insight here is to take both $F_X$ and $G_Y$ as pre-trained unimodal encoders which we keep frozen throughout, and treat our V-L adapters $A_X$ and $A_Y$ as learnable heads for multimodal alignment. Therefore, we can define our learning objective using the InfoNCE loss function as follows:

$$\mathcal{L}_{\text{NCE}} = -\log \frac{\exp(\text{sim}(f_X(\mathcal{X}_i), f_Y(\mathcal{Y}_i))/\tau)}{\sum_{j=1}^{N} \exp(\text{sim}(f_X(\mathcal{X}_i), f_Y(\mathcal{Y}_j))/\tau)} \tag{1}$$

where $\text{sim}(\cdot, \cdot)$ denotes the similarity function, $\tau$ is the temperature parameter, and $N$ is the number of all samples. Where $\mathcal{X}_i$ and $\mathcal{Y}_i$ are positive pairs.

**Motivation:** Scaling up multimodal models boosts performance but incurs substantial computational costs, especially when jointly training networks $F_X$ and $G_Y$, which rapidly increases memory and compute demands. Besides, acquiring high-quality paired data is costly, while high-quality unimodal data is more readily available and can provide rich supervision through self-supervised learning. To address these challenges, we aim to design a computationally efficient, V-L alignment adapters $A_X$ and $A_Y$ ,that reduces reliance on paired data by leveraging unimodal signals, while allowing independent updates to visual and textual components with minimal retraining.

Additionally, the existing image-text matching datasets, mostly sourced from the internet (Figure 4 Upper Figure), generally have low pairing quality. In contrastive learning that emphasize the quality of positive sample pairings, ambiguously matched positive pairs can lead to excessive model confidence, despite the pairs not being correctly matched. Therefore, there is a need for a method to adjust the matching degree of positive pairs and mitigate model overconfidence in these samples.

### 3.2 OVERVIEW

We introduce Align-VL, a multimodal alignment method that adjusts data matching by simulating input uncertainty and moderating model overconfidence in ambiguous positive samples, operating on the latent spaces $\mathcal{U}_\mathcal{X}$ and $\mathcal{U}_\mathcal{Y}$ derived from pre-trained unimodal encoders. 1) The method initially employs unimodal encoders to encode V-L modalities into intermediate latent spaces. 2) Subsequently, it utilizes enhanced features based on Fusemix in the Align-VL latent spaces, incorporating Random Perturbation to adjust data matching by simulating input uncertainty. 3) After training through VL-Adapters, Embedding Smoothing is applied, aiming to smooth the model's prediction of output distributions, reduce overconfidence in positive pairs, and enhance the smoothness of predictions on uncertain samples. 4) Finally, the smoothed embeddings are used in contrastive learning training, facilitating the learning of two networks, $A_X$ and $A_Y$, as shown in Figure 3.

Align-VL utilizes the existing semantics encoded by unimodal encoders, reducing the reliance on extensive real paired data and simplifying computational requirements. It effectively mitigates the issue of model overconfidence, making the model more "modest" and robust, thereby optimizing multimodal alignment, learning efficiency, and generalization capabilities.

## 3.3 RANDOM PERTURBATION HELPS ENHANCE FEATURE ROBUSTNESS.

We introduce random perturbation in the Align-VL latent space to adjust data matching by simulating input uncertainty. Given a visual input $\mathcal{X}$ and a textual input $\mathcal{Y}$, their respective embeddings $\mathbf{z}_v = F_X(\mathcal{X})$ and $\mathbf{z}_t = G_Y(\mathcal{Y})$ are obtained from the visual encoder $F_X$ and the text encoder $G_Y$. In training phase, we apply Gaussian noise Willett et al. (2000) to these embeddings by sampling noise vectors $\mathbf{n}_v \sim \mathcal{N}(0, \sigma^2)$ and $\mathbf{n}_t \sim \mathcal{N}(0, \sigma^2)$, resulting in perturbed embeddings. The perturbed embeddings are defined as:

$$\begin{aligned}
\tilde{\mathbf{z}}_v &= \mathbf{z}_v + \sigma \cdot \epsilon_v, \quad \epsilon_v \sim \mathcal{N}(0, I) \\
\tilde{\mathbf{z}}_t &= \mathbf{z}_t + \sigma \cdot \epsilon_t, \quad \epsilon_t \sim \mathcal{N}(0, I)
\end{aligned} \tag{2}$$

where $\sigma$ represents the noise level, and $\epsilon_v$ and $\epsilon_t$ are random noise vectors sampled from the standard normal distribution. This noise is added only during the training phase to prevent the model from becoming overconfident in its learned representations, ensuring that it explores a wider range of possible embeddings and becomes more robust to small variations in the input data. By perturbing the embeddings during training, the model is forced to learn representations that are invariant to these perturbations, thus improving generalization (See the Appendix subsection A.3 for more analysis).

To show the regularizing effect of Gaussian noise, we examine the variance it introduces, noting that the expected value of the perturbed embedding remains unchanged from the original.

$$\mathbb{E}[\tilde{\mathbf{z}}_v] = \mathbb{E}[\mathbf{z}_v + \sigma \epsilon_v] = \mathbf{z}_v \tag{3}$$

However, the variance of the perturbed embedding increases due to the Gaussian noise, which adds a regularization effect to the model. The variance of the perturbed embedding is given by:

$$\mathrm{Var}[\tilde{\mathbf{z}}_v] = \mathrm{Var}[\mathbf{z}_v + \sigma \epsilon_v] = \sigma^2 \cdot I \tag{4}$$

Thus, the total variance of the embeddings with Gaussian noise becomes:

$$\mathrm{Var}_{\text{total}} = \mathrm{Var}[\mathbf{z}_v] + \sigma^2 \tag{5}$$

Introducing the noise adds variance, helping the model avoid overfitting by promoting smoother decision boundaries and better generalization. This encourages the learning of robust features that perform well on unseen data by broadening the explored feature space during training. For inference, the noise is removed to ensure accurate predictions, returning embeddings to their original state.

$$\mathbf{z}_v = A_X(\mathbf{x}_v), \quad \mathbf{z}_t = A_Y(\mathbf{x}_t) \tag{6}$$

## 3.4 EMBEDDING SMOOTHING HELPS MODERATE MODEL OVERCONFIDENCE

**Embedding Smoothing and Modified Loss Function:** Inspired by label smoothing Gong et al. (2024); Müller et al. (2019), to reduce the model's reliance on hard embeddings and enhance generalization, we design Embedding Smoothing (ES) into our Align-VL framework. In this context, each example within a batch is treated as a unique class, making the number of classes $N$ equivalent to the batch size. For a given target example $y$, the smoothed target distribution is defined as:

$$\tilde{y}_i = \begin{cases} 1 - \alpha, & \text{if } i = y \\ \frac{\alpha}{N-1}, & \text{if } i \neq y \end{cases} \tag{7}$$

where $\alpha \in (0, 1)$ is the smoothing parameter, $N$ is the batch size, and $i$ and $y$ are indices of examples within the batch. ES assigns a small non-zero probability to all other examples in the batch, thereby preventing the model from becoming overconfident and improving its ability to generalize.

We incorporate these smoothed targets into a symmetric contrastive loss function. The perturbed embeddings $\tilde{\mathbf{z}}_v$ and $\tilde{\mathbf{z}}_t$ from the two modalities (image and text) are projected using their respective V-L Adapters $A_X$ and $A_Y$. The loss function with Embedding Smoothing is defined as:

$$\mathcal{L}_{\text{sym}}^{\text{EmbedSmooth}} = \frac{1}{2} \left( \mathcal{L}\left( A_X(\tilde{\mathbf{z}}_v), \tilde{Y}; \tilde{\mathbf{z}}_t \right) + \mathcal{L}\left( A_Y(\tilde{\mathbf{z}}_t), \tilde{Y}; \tilde{\mathbf{z}}_v \right) \right) \tag{8}$$

The loss $\mathcal{L}$ is computed using the Kullback-Leibler divergence between the smoothed target distributions and the model's predicted probabilities:

$$\mathcal{L}\left( A_X(\tilde{\mathbf{z}}_v), \tilde{Y}; \tilde{\mathbf{z}}_t \right) = \text{KL}\left( \tilde{y} \,\middle\|\, \text{Softmax}\left( \frac{\text{sim}\left( A_X(\tilde{\mathbf{z}}_v), A_Y(\tilde{\mathbf{z}}_t) \right)}{\tau} \right) \right)$$
$$\mathcal{L}\left( A_Y(\tilde{\mathbf{z}}_t), \tilde{Y}; \tilde{\mathbf{z}}_v \right) = \text{KL}\left( \tilde{y} \,\middle\|\, \text{Softmax}\left( \frac{\text{sim}\left( A_Y(\tilde{\mathbf{z}}_t), A_X(\tilde{\mathbf{z}}_v) \right)}{\tau} \right) \right) \tag{9}$$

where $\text{sim}(\cdot, \cdot)$ denotes a cosine similarity measure. $\tau$ is a hyperparameter that controls the concentration of the distribution. $\text{Softmax}(\cdot)$ converts similarity scores into a probability distribution over the batch. By utilizing ES, the model is encouraged to produce output distributions that are less peaked and more spread out, which helps in preventing overfitting. The symmetric loss function ensures equal contributions from both modalities during the learning process.

**Theoretical Analysis:** ES increases the entropy of target distributions by assigning non-zero probabilities to all classes, reducing model overconfidence and reliance on specific training examples, thereby enhancing robustness and generalization. Theoretically, it serves as regularization, preventing excessive focus on any single class and promoting better generalization to unseen data. The increase in entropy of the target distribution can be quantified. For the smoothed target distribution $\tilde{y}$, the entropy is:

$$H(\tilde{y}) = -\left( (1 - \alpha) \log(1 - \alpha) + (N - 1)\left( \frac{\alpha}{N - 1} \log \frac{\alpha}{N - 1} \right) \right) \tag{10}$$

Higher entropy in the target distribution smooths gradients in training, serving as regularization to prevent overfitting. ES also improves model calibration by using KL divergence to discourage overconfidence, promoting even probability distribution across positive and negative samples (refer to Appendix subsection A.2 for detailed analysis).

## 4 EXPERIMENTS

**Baselines and Metrics:** We conduct comparisons with several multimodal alignment methods: Fusemix Vouitsis et al. (2024), CLIP Radford et al. (2021), LIT Zhai et al. (2022), and 3T Kossen et al. (2024), as shown in Table 1 and Table 2. It's important to note that large-scale image-text datasets used by models like CLIP, LIT, and 3T, ranging from 400 million to 5 billion pairs, are mostly sourced from the internet and not publicly available, with high computational costs making them impractical for limited-resource scenarios. Therefore, Fusemix, detailed in Table 1, offers a more applicable comparison with Align-VL. To assess the performance of multimodal alignment for all considered methods, we use metrics such as R@1, R@5, and R@10. R@1 denotes Recall@1 for either text-to-image or image-to-text, R@5 indicates Recall@5, and R@10 stands for Recall@10, applicable to both text-to-image and image-to-text retrieval scenarios Vouitsis et al. (2024).

**Experimental Setup:** To minimize computational demands, all our experiments are conducted on a single 24GB NVIDIA 3090 GPU. We pre-compute latents from pre-trained unimodal encoders, which are then discarded, extracting latents for each modality sequentially to avoid loading more than one encoder at a time. For consistency and fair comparison, we use the same unimodal encoders as Fusemix. V-L adapters are parameterized as lightweight MLPs featuring an inverted bottleneck architecture, inspired by previous studies Lin et al. (2015); Tolstikhin et al. (2021); Bachmann et al. (2023). Each MLP incorporates residual blocks and a default final projection layer with a dimension of 512, embedding each modality into a shared latent space. For the image encoder, we consider

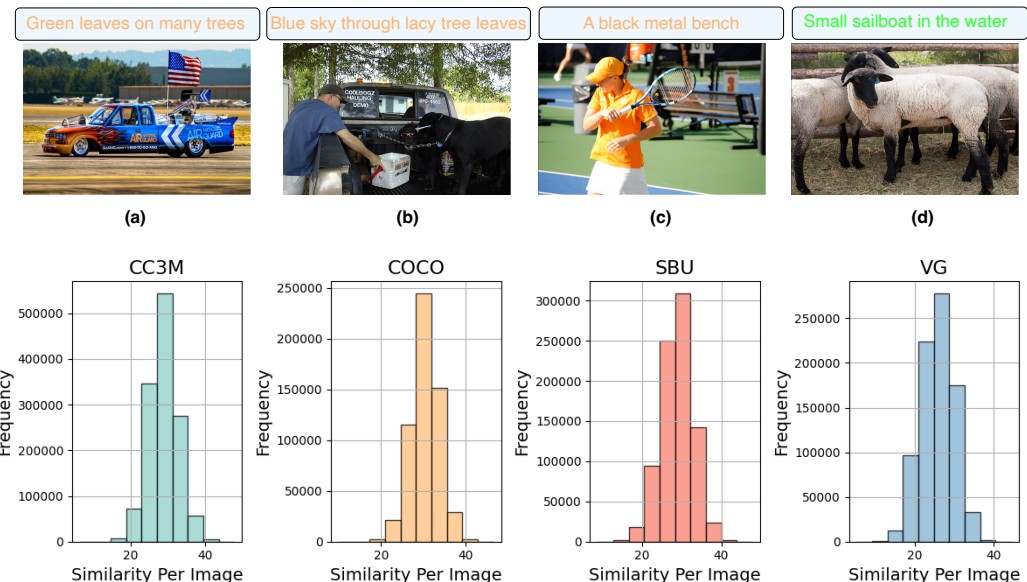

Figure 4: The upper figure mentions "ambiguous samples" in datasets, including partially matched samples (a, b, c) and completely unmatched samples (d). During contrastive learning training, it can lead to positive pairs not being truly positive. The lower figure use CLIP to compute similarity for four datasets (For details, see Table 6), revealing generally moderate alignment between images and texts (mostly ranging from 20-40%). Notably, the COCO dataset shows higher alignment (indicative of superior data quality), which correlates with relatively higher performance metrics.

DINOv2 Oquab et al. (2023), and for the text side, we select text encoder with demonstrably semantic latent spaces, specifically BGE Xiao et al. (2023). We highlight that since our V-L adapters are operating on low-dimensional latents, the computational cost to train them is minimal, and despite training on a single GPU, we can use large batch sizes (up to $Batch = 10$K on 3090 GPU), which has been shown to benefit contrastive learning Wu et al. (2018); Tian et al. (2020); He et al. (2020). The smoothing parameter $\alpha$ is set to 0.1, and the Gaussian noise level $\sigma$ is set to 0.01.

**Training Datasets.** To evaluate the effectiveness of the Align-VL method for the task of modality alignment, we conducted extensive comparative experiments against SoTA methods across various datasets. following previous works Chen et al. (2020); Li et al. (2021; 2022; 2023), These datasets include COCO (human-annotated) Lin et al. (2014b), Visual Genome (VG) (human-annotated) Krishna et al. (2017a), SBU (web datasets) Ordonez et al. (2011b), and Conceptual Captions 3M (CC3M, web datasets) Sharma et al. (2018b). Table 6 provides detailed information about these four datasets. It is noteworthy that the original CC3M dataset, consisting of images stored as internet URLs, currently has only 1.5 million data pairs available. Using a single NVIDIA 3090 GPU for training, Align-VL achieved SoTA performance across datasets of comparable sizes. Detailed results and analyses are presented in the subsequent sections.

## 4.1 RESULTS & DISCUSSIONS

**Benchmark Comparison:** As demonstrated in our experiments in Table 1, Align-VL consistently outperforms Fusemix across various dataset sizes and configurations in image-text retrieval tasks. For example, on the COCO dataset ($\approx 560K$ pairs), Align-VL achieves a significant improvement in text-to-image retrieval, with a 3% higher R@1 score compared to Fusemix. On SBU dataset ($\approx 840K$ pairs), Align-VL surpasses Fusemix by over 4.48% in R@1 for text-to-image tasks. Furthermore, on a combined training configuration of four datasets (VG+COCO+SBU+CC3M, totaling $\approx 3.5M$ pairs), Align-VL achieves a significant improvement of over 1.44% in R@1 for text-to-image retrieval. These results highlight Align-VL's robust generalization capabilities and superior performance over the SoTA method.

The Align-VL demonstrates a significant advantage in achieving high performance with substantially lower training costs. Compared to training datasets of similar size, Align-VL outperforms

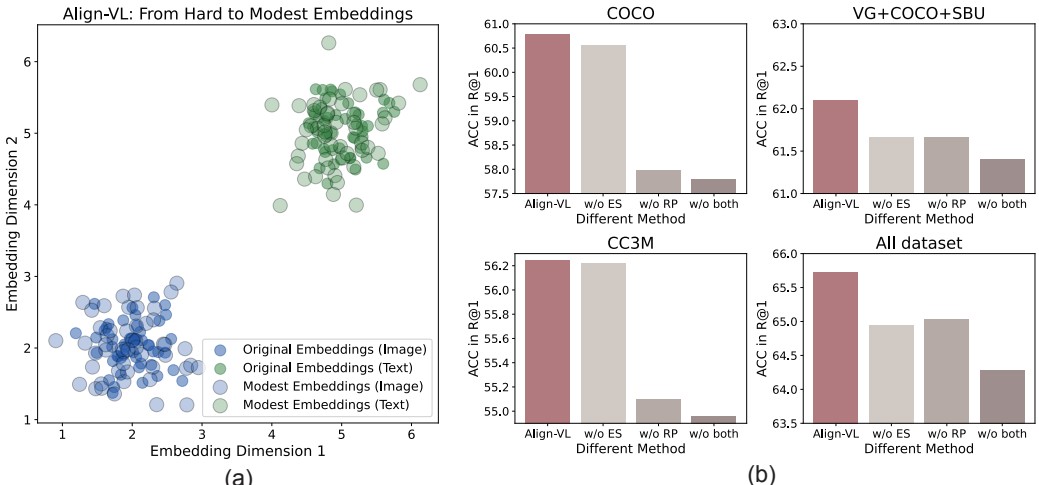

(a)                                            (b)

Figure 5: (a) In the original distribution, clusters of image-text pairs are tightly packed, showing high model confidence. After implementing the "Be Modest" Align-VL approach, the embeddings become more dispersed, indicating reduced confidence in individual labels and a more robust, softer probability distribution that enhances model generalization. (b) The ablation study of two techniques (Random Perturbation (RP) and Embedding Smoothing (ES)).

Table 1: The performance of Align-VL and Fusemix Vouitsis et al. (2024) on Flickr30K test set. By evaluating on multiple datasets with varying sizes and complexities, we show that our Align-VL model exhibits strong generalization capabilities and achieves SoTA performance on image retrieval tasks. Bold signifies the best.

| Size | Training Dataset | Method | Flickr30K (1K test set) | | | | | |
| | | | text → image | | | image → text | | |
| | | | R@1 | R@5 | R@10 | R@1 | R@5 | R@10 |
| ≈ 560K | COCO | Fusemix (Vouitsis et al., 2024) | 57.80 | 83.38 | 89.54 | 71.60 | 91.10 | 95.00 |
| | | Align-VL (ours) | **60.80** | **84.82** | **90.82** | **72.60** | **93.30** | **95.70** |
| ≈ 820K | VG | Fusemix (Vouitsis et al., 2024) | 51.66 | 79.20 | 86.66 | 66.90 | 90.20 | 95.40 |
| | | Align-VL (ours) | **52.90** | **80.64** | **87.86** | **70.20** | **90.40** | **95.90** |
| ≈ 840K | SBU | Fusemix (Vouitsis et al., 2024) | 43.32 | 72.48 | 81.36 | 61.60 | 86.30 | 92.10 |
| | | Align-VL (ours) | **47.80** | **75.94** | **84.18** | **62.10** | **87.00** | **92.30** |
| 1.38M | VG+COCO | Fusemix (Vouitsis et al., 2024) | **61.56** | 86.00 | 91.40 | 76.40 | 93.60 | 97.10 |
| | | Align-VL (ours) | 61.54 | **86.16** | **91.72** | **77.20** | **94.40** | **97.80** |
| 2M | VG+COCO+SBU | Fusemix (Vouitsis et al., 2024) | 61.40 | 84.82 | 90.46 | 77.20 | 94.40 | 97.20 |
| | | Align-VL (ours) | **62.10** | **85.52** | **90.76** | **77.80** | **94.60** | **97.70** |
| 3.5M | VG+COCO+SBU+CC3M | Fusemix (Vouitsis et al., 2024) | 64.28 | 87.60 | 91.86 | 81.20 | **96.40** | 98.20 |
| | | Align-VL (ours) | **65.72** | **87.82** | **92.50** | **81.60** | 96.00 | **98.30** |

both CLIP and Fusemix, achieving an R@1 improvement of 11.42% in text-to-image retrieval and 14.2% in image-to-text retrieval over CLIP (Table 2). Additionally, it surpasses Fusemix with improvements of 5.82% and 7.2% in both tasks, respectively, as detailed in Table 2. Furthermore, despite using significantly smaller training data (3.5M pairs), Align-VL performs competitively against much larger datasets used by CLIP (400M) and LIT (4B), trailing by only 0.78% in text-to-image and 2.30% in image-to-text retrieval against LIT. This demonstrates the effectiveness and efficiency of Align-VL. We anticipate that with further increases in training data, Align-VL has the potential to surpass these SoTA methods, further underscoring its usability.

**Efficiency in Dataset Quality:** To evaluate the dataset quality, we utilized the CLIP-ViT/B-32 model Radford et al. (2021) to compute the similarity between images and texts across four datasets, as shown in Table 6. This analysis reveals weak associations and low data quality between images and texts in exisiting datasets. For the CC3M, 93.5% of the image-text pairs have a similarity of less than 35, indicating a low level of alignment between the images and texts in the dataset. Similar trends are observed in the COCO, SBU, and VG datasets, where 91.9%, 94.4%, and 99.1% of image-text pairs, respectively, have a similarity of less than 35 (See Figure 4 lower figure). This reflects the prevalence of ambiguous or weakly paired image-text samples across these datasets. It's noteworthy

that COCO exhibits a noticeably higher degree of pairing among these datasets, resulting in higher performance metrics for alignment models trained on it, underscoring the importance of data quality (Table 1). In contrastive learning, datasets with poorer quality lead to many ambiguous positive pairs, causing models to become overly confident during training and confused during inference. Therefore, Align-VL becomes particularly crucial in addressing these challenges.

Table 2: The performance of SoTA methods and Align-VL on different training datasets is assessed on the Flickr30K dataset's 1K test set, evaluating text-to-image and image-to-text retrieval accuracy using R@1 scores.

| Size | Method | Flickr30K (1K test set) | |
|---|---|---|---|
| | | text → image | image → text |
| 400M | CLIP (Radford et al., 2021) | 68.70 | 88.00 |
| 4B | LIT (Zhai et al., 2022) | 66.50 | 83.90 |
| 5B | 3T (Kossen et al., 2024) | 72.10 | 87.30 |
| 3M | CLIP (Radford et al., 2021) | 54.30 | 67.40 |
| 3M | Fusemix$_{(D,B)}$ (Vouitsis et al., 2024) | 59.90 | 74.40 |
| 5M | Fusemix$_{(U,E)}$ (Vouitsis et al., 2024) | 64.30 | 80.20 |
| 3.5M | Align-VL (Ours) | **65.72** | **81.60** |

## 4.2 ABLATION STUDY

**Effect of Random Perturbation (RP):** To validate the effectiveness of RP, we conducted ablation experiments for RP. As shown in Table 3, adding RP consistently improves performance across all datasets. On the COCO dataset, adding Random Perturbation (RP) in two settings—with and without Embedding Smoothing (ES)—results in an R@1 score increase from 57.98% to 60.8%, a 2.82% improvement, and from 57.8% to 60.56%, a 2.76% improvement. Similarly, this positive trend in gains from RP is observable across other datasets as well, as shown in Figure 5 (b). These results demonstrate that RP significantly enhances model performance regardless of whether ES techniques are used.

**Effect of Embedding Smoothing (ES):** As shown in Table 3, ES improves the R@1 scores on the SBU dataset by 3.8% without Random Perturbation (RP) and by 1.6% with RP. Similar gains from ES are observed across other datasets, regardless of the RP setting, as shown in Figure 5. ES enhances performance on complex, large-scale datasets by improving generalization across diverse image-text pairs. As a regularization strategy in contrastive learning, it smooths target distributions to mitigate overfitting, thereby boosting model robustness and accuracy on validation datasets.

Table 3: Ablation experiments of different techniques, including Random Perturbation (RP) and Embedding Smoothing (ES), are conducted across various datasets, measuring text-to-image retrieval performance (R@1).

| RP | ES | Training Dataset | | | |
|---|---|---|---|---|---|
| | | COCO | SUB | VG+COCO+SBU | VG+COCO+SBU+CC3M |
| | | 57.80 | 43.32 | 61.40 | 64.28 |
| | ✔ | 57.98 | 47.12 | 61.66 | 64.94 |
| ✔ | | 60.56 | 46.20 | 61.66 | 65.04 |
| ✔ | ✔ | **60.80** | **47.80** | **62.10** | **65.72** |

**Effect of Smoothing Parameter $\alpha$ in ES:** As shown in Table 4, default parameter ES consistently outperforms Dynamic $\alpha$ of ES (with $\alpha$ decreasing over training), demonstrating the default $\alpha$'s effectiveness. On the COCO dataset, default ES achieves a marginal but notable 0.34% improvement in R@1 scores for text-to-image retrieval, while the improvement in image-to-text retrieval is more substantial, with a 0.8% increase. These results are consistent across other datasets, where default $\alpha$ ES consistently delivers performance enhancements. While dynamic ES offers certain benefits, default ES is more effective in optimizing retrieval tasks, particularly by adapting to diverse datasets and handling complex data more efficiently. Both dynamic and default smoothing parameters significantly outperform models without ES, underscoring ES's effectiveness.

Table 4: Ablation Study of Different ES Types in Various Datasets (R@1). This study compares the performance of different types of Embedding Smoothing (ES) across several datasets. We use ✗ to denote without ES, ◯ for dynamic $\alpha$ in ES, and ✓ for default $\alpha$ in ES.

| | Different Type of ES | | | | | |
| | text → image | | | image → text | | |
| Dataset | ✗ | ◯ | ✓ | ✗ | ◯ | ✓ |
|---|---|---|---|---|---|---|
| COCO | 57.80 | 60.22 | **60.56** | 71.60 | 72.70 | **73.50** |
| VG | 51.66 | 52.50 | **53.80** | 66.90 | 68.10 | **69.20** |
| SUB | 43.32 | 45.46 | **47.12** | 61.60 | 63.60 | **62.80** |
| VG+COCO | 61.56 | 61.98 | **62.24** | 76.40 | 77.60 | **77.20** |
| VG+COCO+SUB | 61.40 | 61.32 | **61.66** | 77.20 | 77.90 | **78.40** |
| VG+COCO+SUB+CC3M | 64.28 | 65.14 | 64.94 | 81.20 | 81.50 | **82.50** |

**Impact of Dataset Quality:** Smaller, human-annotated datasets like COCO ($\approx 560K$) can significantly outperform larger, web-sourced ones like the SBU ($\approx 840K$) in retrieval tasks, as reflected in their R@1 scores (57.80% vs 43.32% in text-image R@1, Table 5). Similar observations are made when comparing combined datasets of VG+COCO+SBU with CC3M. This highlights that careful curation of datasets often leads to better model performance and generalization than merely increasing dataset size. However, curating large-scale datasets is impractical, making the capability of Align-VL to adjust the degree of image-text match within the dataset (potentially enhancing data quality) particularly valuable. As shown in Figure 5 (a), after processing with the Align-VL algorithm, embeddings shift from "hard" to "modest" and become smoother, indicating reduced confidence in individual samples and a more dispersed, robust probability distribution that enhances model generalization. Align-VL potentially enhances dataset quality and the subsequent improvement in performance metrics across all datasets underscores its effectiveness.

Table 5: Analysis of the Impact of Dataset Quality and Size with Align-VL. This analysis compares the effects of varying dataset quality and size on the Align-VL model's performance. The smaller datasets of higher quality outperform larger but lower-quality datasets, underscoring the importance of data quality.

| Size | Dataset | text → image | | | image → text | | |
| | | R@1 | R@5 | R@10 | R@1 | R@5 | R@10 |
|---|---|---|---|---|---|---|---|
| $\approx 3M$ | CC3M | 59.90 | 86.40 | 91.60 | 74.70 | 94.00 | 97.40 |
| $\approx 2M$ | VG+COCO+SBU | 61.40 | 84.82 | 90.46 | 77.20 | 94.40 | 97.20 |
| $\approx 820K$ | VG | 51.66 | 79.20 | 86.66 | 66.90 | 90.20 | 95.40 |
| $\approx 840K$ | SUB | 43.32 | 72.48 | 81.36 | 61.60 | 86.30 | 92.10 |
| $\approx 560K$ | COCO | 57.80 | 83.38 | 89.54 | 71.60 | 91.10 | 95.00 |

## 5 CONCLUSION

In this work, we propose a multimodal alignment framework Align-VL that promotes modesty in model predictions while being computationally and data-efficient. Align-VL is a straightforward yet effective method that reduces model overconfidence and enhances the intrinsic connections of paired data in the latent space. It effectively leverages guidance from pretrained visual and text unimodal encoders. Notably, Align-VL excels on datasets with lower data quality (image-text match level) and also enhances performance on datasets with higher match levels. Validated across multiple datasets, Align-VL has consistently demonstrated its robust capability to align V-L models better. For future developments, Align-VL could incorporate data quality assessments to dynamically adjust model confidence, applying stricter constraints on lower quality data and more lenient ones on higher quality data to effectively manage model overconfidence.

## LIMITATIONS

Although Align-VL achieved performance improvements across all datasets and their combinations, it is notable that the gains are more significant for datasets of poorer quality, while the enhancements are more modest for relatively better datasets. We are unable to test Align-VL on larger datasets like the 400M pairs used in CLIP, as it is not publicly available. Consequently, it is difficult to ascertain Align-VL's performance benefits for extremely large-scale datasets. Besides, it remains unclear how much Align-VL benefits datasets of very high quality, where image-text pairs are perfectly matched.

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

# A  APPENDIX

In this section, we present additional implementation details, experiment results, theoretical analysis, pseudo code and supplements. The content structure is outlined as follows:

## A.1  ASSESSING THE MATCH QUALITY OF IMAGE-TEXT DATASETS WITH CLIP

### A.1.1  SIMILARITY CALCULATION

In the context of contrastive learning models such as CLIP, the similarity produced for a given image-text pair is closely related to the cosine similarity of their respective embeddings, modulated by a temperature scaling factor $\tau$. Specifically, let $Similarity(i, j)$ denote the similarity score for image $i$ and text $j$. This score can be mathematically expressed as:

$$\mathrm{S}imilarity(i, j) = \frac{\text{image\_embedding}(i) \cdot \text{text\_embedding}(j)}{\|\text{image\_embedding}(i)\|\|\text{text\_embedding}(j)\|} \times \frac{1}{\tau} \tag{11}$$

where $\tau$ is the temperature coefficient, which is set to $0.01$. Consequently, when multiplied by the temperature coefficient $\frac{1}{\tau}$, the similarity score will be constrained within the new range.

### A.1.2  SIMILARITY IN FOUR DATASETS

To assess the match quality of image-text datasets in the main text, we conducted similarity calculation experiments on four widely used image-text datasets: COCO, CC3M, SBU, and VG. These datasets cover a diverse range of image content, from everyday objects to complex scenes, and vary in terms of size and annotation quality. Table 6 presents a summary of these datasets, detailing the number of image-text pairs in each and the ranges of similarity scores calculated for different image-text pairings.

In a similarity analysis, the COCO dataset exhibited the highest mean similarity per image at 30.48, indicating a strong and consistent association between images and text. The CC3M and SBU datasets demonstrated similar mean similarities of 28.80, reflecting comparable levels of alignment quality. This highlights how dataset structure and complexity significantly influence model performance in multimodal tasks, as shown in Figure 4 (b). Table 6 shows the distribution of similarity for Image-Text pairs across four datasets. It can be observed that the values are concentrated around the 30 range, indicating that when CLIP is used as a scoring model, the resulting similarity tends to cluster within a relatively modest range.

Table 6: Similarity Per Image-Text Across Ranges for Different Datasets. The table shows the count of similarity scores within specific similarity ranges for each dataset (CC3M, COCO, SBU, and VG). Each row corresponds to a dataset, and each column represents a range of similarity values.

|       | [5,10) | [10, 15) | [15, 20) | [20, 25) | [25, 30) | [30, 35) | [35, 40) | [40, 45) | [45, 50) |
|-------|--------|----------|----------|----------|----------|----------|----------|----------|----------|
| **CC3M** | 11  | 400  | 14877 | 189355 | 616912 | 408777 | 70836 | 3389 | 32 |
| **COCO** | 1   | 29   | 1187  | 27566  | 216780 | 275238 | 44607 | 1331 | 8  |
| **SBU**  | 15  | 647  | 15268 | 127793 | 365869 | 284130 | 45279 | 1702 | 24 |
| **VG**   | 39  | 3149 | 68441 | 267798 | 342860 | 132259 | 7151  | 77   | 0  |

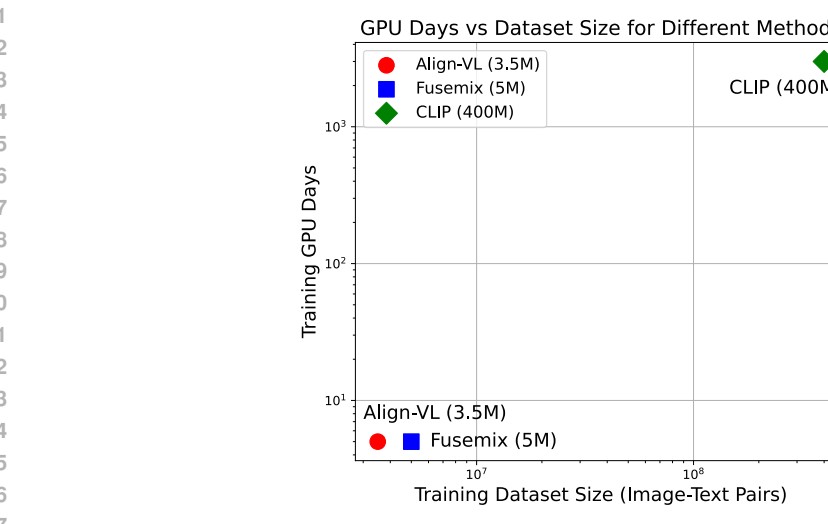

Figure 6: In terms of training efficiency, CLIP requires 3000 GPU days for training on 400 million data pairs, while Align-VL needs only approximately 5 GPU days for 3.5 million data pairs.

## A.2 THEORETICAL ANALYSIS FOR EMBEDDING SMOOTHING

Embedding Smoothing effectively increases the entropy of the target distributions by assigning non-zero probabilities to all classes (examples in the batch). This reduction in confidence prevents the model from becoming overly reliant on specific training examples, thereby enhancing its robustness. The inclusion of the smoothing parameter $\alpha$ allows for control over the degree of smoothing applied, enabling a balance between model confidence and generalization ability. To provide a theoretical understanding of how Embedding Smoothing improves generalization, we analyze its impact on the loss function and the model's predictions.

In standard contrastive learning without smoothing, the loss for a positive pair is:

$$\mathcal{L}_{\text{pos}} = -\log \frac{\exp\left(\text{sim}\left(A_X(z_x), A_Y(z_y)\right)/\tau\right)}{\sum_{j=1}^{N} \exp\left(\text{sim}\left(A_X(z_x), A_Y(z_j)\right)/\tau\right)} \tag{12}$$

This loss encourages the model to maximize the similarity between positive pairs and minimize it between negative pairs. However, it can lead to overconfident predictions, as the model focuses heavily on the positive pair. With Embedding Smoothing, the loss incorporates the smoothed target distribution $\tilde{y}$, and the KL divergence becomes:

$$\mathcal{L}\left(A_X(z_x), \tilde{Y}; Z_Y\right) = -\sum_{i=1}^{N} \tilde{y}_i \log p_i \tag{13}$$

where $p_i$ is the predicted probability for the $i$-th example in the batch:

$$p_i = \frac{\exp\left(\text{sim}\left(A_X(z_x), A_Y(z_i)\right)/\tau\right)}{\sum_{j=1}^{N} \exp\left(\text{sim}\left(A_X(z_x), A_Y(z_j)\right)/\tau\right)} \tag{14}$$

By assigning non-zero probabilities $\tilde{y}_i$ to all classes, the loss function penalizes the model not only for the positive pair but also for negative pairs, albeit to a lesser extent. This encourages the model to produce a probability distribution that is more uniform and less confident.

Analyzing from the perspective of information entropy: The entropy $H(\tilde{y})$ of the smoothed target distribution is higher than that of a one-hot distribution. The entropy of $\tilde{y}$ is:

$$H(\tilde{y}) = -\left((1-\alpha)\log(1-\alpha) + (N-1)\left(\frac{\alpha}{N-1}\log\frac{\alpha}{N-1}\right)\right) \tag{15}$$

Higher entropy in the target distribution leads to smoother gradients during training, which can prevent the model from fitting noise in the training data. This smoothing effect acts as a form of regularization, reducing overfitting.

**Reduction in Overconfident Predictions:** Embedding Smoothing reduces the Kullback-Leibler divergence between the predicted distribution $p$ and the uniform distribution $u$, where $u_i = \frac{1}{N}$:

$$\text{KL}(u\|p) = \sum_{i=1}^{N} u_i \log\frac{u_i}{p_i} \tag{16}$$

By making $p$ closer to $\tilde{y}$, which has higher entropy, the model's predictions become less confident. This can be beneficial because overconfident predictions on training data often lead to poor generalization on unseen data.

**Connection to Label Smoothing Theory:** Embedding Smoothing in our context is analogous to label smoothing in classification tasks. Previous works have shown that label smoothing has the following effects: 1). Margin Maximization: It implicitly increases the decision margin between classes, which can improve generalization. 2). Penalization of Confident Wrong Predictions: By smoothing the targets, the loss function penalizes overconfident incorrect predictions more heavily.

Besides, consider the gradient of the loss with respect to the logits $z$:

$$\frac{\partial\mathcal{L}}{\partial z_i} = p_i - \tilde{y}_i \tag{17}$$

When using Embedding Smoothing, $\tilde{y}_i$ is never exactly 0 or 1. This means that the gradients are non-zero for all classes, encouraging the model to adjust its predictions across all examples in the batch. This leads to more generalized feature representations. By incorporating Embedding Smoothing into the loss function, we introduce a regularization effect that enhances the model's generalization capabilities. The smoothing parameter $\alpha$ provides a mechanism to control this effect, allowing for a trade-off between fitting the training data and maintaining robustness to unseen data. This theoretical understanding aligns with our experimental observations, where models trained with Embedding Smoothing demonstrate improved performance on validation datasets.

### A.3 THEORETICAL ANALYSIS FOR RANDOM PERTURBATION

**Why Choose Gaussian noise?:** This paper introduces Gaussian noise as a perturbation in Align-VL for several reasons: 1). Well-Defined Mathematical Properties: Gaussian distribution exhibits continuous and smooth probability density functions across the real number line, facilitating theoretical analysis and calculations. 2). Zero-Mean Symmetry: By choosing a Gaussian distribution with a mean of zero, the added noise is symmetrically balanced around zero, introducing no systematic bias and only increasing the variance, thereby preserving the expected value of embeddings. 3). Adjustable Perturbation Intensity: The standard deviation of the Gaussian distribution can be precisely controlled, allowing for careful calibration of noise intensity. This flexibility is crucial for introducing an appropriate level of uncertainty to enhance model robustness. 4). Alignment with Natural Phenomena: According to the Central Limit Theorem, the sum of many independent random variables tends toward a Gaussian distribution. Thus, Gaussian noise effectively simulates random disturbances or measurement errors prevalent in natural and engineering contexts. 5). Facilitation of Optimization and Training: In deep learning, incorporating Gaussian noise helps to smooth the loss function landscape, avoiding local minima and promoting more effective training processes.

In summary, choosing Gaussian noise as the source of perturbation provides theoretical soundness and practical convenience, aiding in the development of more robust feature representations, preventing overfitting, and enhancing the generalization capabilities of models.

**Loss Function with Perturbed Embeddings:** The perturbed embeddings are used in the training loss, specifically in contrastive learning with the InfoNCE loss. The objective is to maximize the similarity between the noisy visual and textual embeddings:

$$\mathcal{L}_{\text{NCE}} = -\log \frac{\exp(\text{sim}(\tilde{\mathbf{z}}_v, \tilde{\mathbf{z}}_t)/\tau)}{\sum_{j=1}^{N} \exp(\text{sim}(\tilde{\mathbf{z}}_v, \tilde{\mathbf{z}}_t^j)/\tau)} \tag{18}$$

where $\text{sim}(\cdot, \cdot)$ represents a similarity function, $\tau$ is a temperature parameter, and $N$ is the number of negative samples. This formulation ensures that the model learns representations that are invariant to noise, thus improving generalization.

**Noise and Regularization Effect:** To understand the impact of Gaussian noise on regularization, we first look at the expected value of the perturbed embeddings. Since the noise is zero-mean, the expected value of the perturbed embeddings is identical to the original embeddings:

$$\mathbb{E}[\tilde{\mathbf{z}}_v] = \mathbb{E}[\mathbf{z}_v + \sigma \epsilon_v] = \mathbf{z}_v \tag{19}$$

However, the variance of the perturbed embeddings increases due to the added noise. The variance of the perturbed embeddings can be calculated as:

$$\text{Var}[\tilde{\mathbf{z}}_v] = \text{Var}[\mathbf{z}_v + \sigma \epsilon_v] = \text{Var}[\mathbf{z}_v] + \sigma^2 \text{Var}[\epsilon_v] \tag{20}$$

Given that $\text{Var}[\epsilon_v] = I$, where $I$ is the identity matrix, the total variance of the perturbed embeddings becomes:

$$\text{Var}_{\text{total}} = \text{Var}[\mathbf{z}_v] + \sigma^2 I \tag{21}$$

The additional term $\sigma^2 I$ acts as a regularizer, which spreads out the embeddings and prevents the model from becoming overconfident in its predictions.

**Minimizing the Generalization Error:** The added noise effectively smooths the decision boundary of the model, which reduces overfitting. By introducing noise, we minimize the generalization error. Assuming the model's prediction function is $f(\mathbf{z})$ and the true function is $f^*(\mathbf{z})$, the goal is to minimize the expected generalization error:

$$\mathbb{E}_{\mathbf{z}}[(f(\mathbf{z}) - f^*(\mathbf{z}))^2] \tag{22}$$

With noise perturbation, the variance in the embeddings increases, which forces the model to learn smoother decision boundaries. The regularization effect introduced by the noise helps bind the generalization error:

$$\mathbb{E}_{\mathbf{z}}[(f(\mathbf{z}) - f^*(\mathbf{z}))^2] \leq \text{Var}_{\text{total}} = \text{Var}[\mathbf{z}_v] + \sigma^2 \tag{23}$$

Thus, the noise helps control the generalization error by ensuring that the model does not overfit to specific features of the training data, which is especially important in cases where the training data contains noise or is limited in size.

**Noise-Induced Gradient Regularization:** We can also analyze the effect of noise on the gradient of the loss function. Given a loss function $\mathcal{L}(\mathbf{z}_v, \mathbf{z}_t)$, the gradient with respect to the perturbed embeddings can be expressed as:

$$\nabla_{\tilde{\mathbf{z}}_v} \mathcal{L}(\tilde{\mathbf{z}}_v, \tilde{\mathbf{z}}_t) = \nabla_{\mathbf{z}_v} \mathcal{L}(\mathbf{z}_v, \mathbf{z}_t) + \sigma \cdot \nabla_{\epsilon_v} \mathcal{L}(\tilde{\mathbf{z}}_v, \tilde{\mathbf{z}}_t) \tag{24}$$

The second term, $\sigma \cdot \nabla_{\epsilon_v} \mathcal{L}(\tilde{\mathbf{z}}_v, \tilde{\mathbf{z}}_t)$, acts as a regularizer that prevents the gradient from becoming too large. The gradient is smoothed by the presence of noise, which further prevents overfitting and encourages the model to learn more generalizable patterns.

**Inference Phase:** During the inference phase, we remove the Gaussian noise to ensure accurate predictions on unseen data. The embeddings revert to their original clean form:

$$\mathbf{z}_v = F(\mathbf{x}_v), \quad \mathbf{z}_t = G(\mathbf{x}_t) \tag{25}$$

Without the added noise, the model makes precise predictions based on the robust features it learned during training. Therefore, by adding Gaussian noise to the embeddings during training, we introduce a form of regularization that improves the generalization ability of the model. The noise prevents overfitting by increasing the variance of the embeddings, ensuring that the model learns smoother decision boundaries. This leads to better performance on unseen data and helps minimize the generalization error. The noise-induced gradient regularization further contributes to preventing the model from overfitting to the training data, making it more robust in real-world applications.

## A.4   IMPLEMENTATION DETAILS

For all experiments, we use the AdamW Loshchilov & Hutter (2018) optimizer during training. We perform learning rate warmup by linearly increasing the learning rate from $10^{-6}$ to $10^{-3}$. We then decay the learning rate using a cosine schedule Loshchilov & Hutter (2016). We use a depth of 4 for both V-L adapters which we train for 500 epochs with a batch size of $Batch = 10K$. We set the learning rate as `lr`$= 10^{-3}$ and use weight decay of 0.1 during optimization. The image encoder is DINOv2 ViT-G/14, and for the text side, the text encoder is the BGE large version. To evaluate the effectiveness of the Align-VL method for the task of modality alignment, we conducted extensive comparative experiments against SoTA methods across various datasets. These datasets include COCO Lin et al. (2014a), VG Krishna et al. (2017b), SBU Ordonez et al. (2011a), and CC3M Sharma et al. (2018a). Table 6 provides detailed information about these four datasets. Additionally, we compared different schemes such as Fusemix, CLIP, and LIT. Utilizing a single NVIDIA 3090 GPU for training, Align-VL demonstrated SoTA performance across datasets of varying sizes.

## A.5   PSEUDOCODE OF ALIGN-VL

---

**Algorithm 1:** PyTorch-style pseudocode for Align-VL

---

**Input:** $A_X, A_Y$: learnable V-L adapters;
$Batch$: batch size;
$D_x, D_y$: latent dimensions of unimodal encoders;
$D_s$: latent dimension of shared space;
$\beta$: Mixup Beta distribution hyperparameter;
$t$: learnable temperature parameter;
$\alpha$: smoothing parameter;
$\sigma$: Gaussian noise level.

1 **for** $z_x, z_y$ *in loader* **do**
    // FuseMix
2    Split $z_x$ and $z_y$ into two parts:
3    $z_{x1}, z_{x2} \in \mathbb{R}^{B \times D_x}, z_{y1}, z_{y2} \in \mathbb{R}^{B \times D_y}$;
4    Sample mixing coefficient $\lambda \sim \text{Beta}(\beta, \beta)$;
5    Mix embeddings:
6    $z_x = \lambda z_{x1} + (1 - \lambda)z_{x2}$;
7    $z_y = \lambda z_{y1} + (1 - \lambda)z_{y2}$;
    // Add Gaussian noise perturbation (training only)
8    **if** *training* **then**
9        $z_x = z_x + \sigma \times \mathcal{N}(0, I)$;
10        $z_y = z_y + \sigma \times \mathcal{N}(0, I)$;
11    **end**
    // Project into joint space and normalize
12    $s_x = \text{normalize}(A_X(z_x)) \in \mathbb{R}^{B \times D_s}$;
13    $s_y = \text{normalize}(A_Y(z_y)) \in \mathbb{R}^{B \times D_s}$;
    // Compute pairwise cosine similarities with temperature
14    $\text{logits}_{xy} = (s_x s_y^\top) \times \exp(t) \in \mathbb{R}^{B \times B}$;
15    $\text{logits}_{yx} = (s_y s_x^\top) \times \exp(t) \in \mathbb{R}^{B \times B}$;
    // Embedding Smoothing
16    **if** $\alpha > 0$ **then**
17        Create smoothed targets:
18        $\tilde{Y} = (1 - \alpha) \times I_B + \dfrac{\alpha}{B-1} \times (\mathbf{1} - I_B) \in \mathbb{R}^{B \times B}$;
19        Compute losses:
20        $\text{loss}_{xy} = \text{KLDivLoss}\left(\log \text{softmax}(\text{logits}_{xy}), \tilde{Y}\right)$;
21        $\text{loss}_{yx} = \text{KLDivLoss}\left(\log \text{softmax}(\text{logits}_{yx}), \tilde{Y}\right)$;
22    **end**
23    **else**
        // Standard symmetric alignment loss
24        Labels: $\text{labels} = [0, 1, \ldots, B-1]$;
25        Compute losses:
26        $\text{loss}_{xy} = \text{CrossEntropyLoss}\left(\text{logits}_{xy}, \text{labels}\right)$;
27        $\text{loss}_{yx} = \text{CrossEntropyLoss}\left(\text{logits}_{yx}, \text{labels}\right)$;
28    **end**
29    Compute average loss:
30    $\text{loss} = \dfrac{\text{loss}_{xy} + \text{loss}_{yx}}{2}$;
    // Optimize
31    optimizer.zero_grad();
32    loss.backward();
33    optimizer.step();
34 **end**

---

