# OpenReview forum: "Align-VL: Can Being Modest Help in the Alignment of Vision-Language Models?"
_ICLR.cc/2025/Conference — Submitted to ICLR 2025_

### Official Review · Reviewer_rzGx · 2024-10-30

**Soundness:** 2
**Presentation:** 3
**Contribution:** 2
**Rating:** 3
**Confidence:** 4

**Summary:**

The paper proposes Align-VL, a vision-language alignment method designed to enhance robustness and manage overconfidence in multimodal models. It incorporates two strategies—Random Perturbation and Embedding Smoothing—to handle uncertainties in image-text datasets, especially when data is ambiguously paired. Align-VL operates on the latent spaces of pre-trained unimodal encoders, aiming to enhance generalization and reduce computational costs.

**Strengths:**

- This paper highlights the reliance of current contrastive learning methods on single positive pairs, a valuable observation.
- The paper is well-structured and clearly written, making it easy to follow.

**Weaknesses:**

- In lines 080 to 082, the claim lacks convincing support. The ALIGN model \[1\] has shown through training on billions of noisy image-text pairs that dataset scale can compensate for noise and yield state-of-the-art representations, which contradicts the author’s claim. Additional references or empirical studies would strengthen this argument.
- The proposed methods, Random Perturbation and Embedding Smoothing are existing methods and are often used as training engineering techniques, which limits their technical novelty.
- The experiment in Table 1 only evaluates a single dataset and compares one baseline, which provides limited evidence.
- The data quality evaluation uses pre-trained CLIP, which may not be ideal given CLIP's own biases, making this evaluation approach potentially limited for assessing data quality.

[1] Jia, Chao, et al. "Scaling up visual and vision-language representation learning with noisy text supervision." ICLR, 2021.

**Questions:**

For specific questions, please refer to the points in the "Weaknesses" section.

---

> ### Author Response · Authors · 2024-11-24
>
> >W1: In lines 080 to 082, the claim lacks convincing support. The ALIGN model [1] has shown through training on billions of noisy image-text pairs that dataset scale can compensate for noise and yield state-of-the-art representations, which contradicts the author’s claim. Additional references or empirical studies would strengthen this argument.
>
> ALIGN demonstrates that large-scale datasets can yield strong representations despite noise, which we discuss further below:
>
> 1. ALIGN achieves strong performance with billions of image-text pairs but at high training costs, limiting its applicability in resource-constrained scenarios. Align-VL addresses this by **enhancing noise robustness and reducing data and computational demands**, enabling broader applicability.
>
> 2. ALIGN no longer achieves state-of-the-art performance*, as shown in the table. Align-VL enhances computational and **data efficiency** for modal alignment in **resource-efficient settings**, outperforming state-of-the-art methods on MSCOCO with significant improvements in R@1 scores. Using just 3.5M data, Align-VL surpasses ALIGN’s performance on MS COCO, achieving this with **300 times less data than ALIGN**.
>
> |     | Text->Image | Image->Text | Training Dataset Size |GPU Days |
> | :-: | :-: | :-:  |:-:  |:-:  |
> | CLIP  | 37.8%  | 58.4%  | 400M  | 3000 |
> | **ALIGN** | 45.6% | 48.6% | 1B  | \ |
> | LIT   | 43.6 | 59.5%  | 4B  | \ |
> | **Align-VL**| **45.86%** | **62.86%**  | **3.5M**  |**5**  |
>
> 3. ALIGN relies on large-scale noisy datasets for performance but is heavily impacted in resource-constrained scenarios. Align-VL mitigates noise effects by **simulating uncertainty in embedding spaces** and addressing overconfidence from weakly correlated data, acting as a feature enhancement for alignment. Incorporating Align-VL’s strategies into ALIGN could further boost its performance.
>
> 4. We will add references [1-2] to support the claim that **limited data amplifies the impact of low-quality samples**, leading to model **overconfidence** and incorrect alignment with mismatched pairs.
>
> We will include additional references and experiments in the revision as you suggested.
>
> [1] Wang, Y., Hong, J., Cheraghian, A., Rahman, S., Ahmedt-Aristizabal, D., Petersson, L., & Harandi, M. (2024). Continual test-time domain adaptation via dynamic sample selection. In Proceedings of the IEEE/CVF Winter Conference on Applications of Computer Vision (pp. 1701-1710).
>
> [2] Xu, Y., Ding, J., Zhang, L., & Zhou, S. (2021). Dp-ssl: Towards robust semi-supervised learning with a few labeled samples. Advances in Neural Information Processing Systems, 34, 15895-15907.
>
> >W2: The proposed methods, Random Perturbation and Embedding Smoothing are existing methods and are often used as training engineering techniques, which limits their technical novelty.
>
>
> We further elaborate on the **unique design and contributions** of Random Perturbation and Embedding Smoothing to multimodal alignment tasks:
> We reject this biased judgment, as our contributions are clearly outlined below:
>
> 1. Previous methods apply random perturbation by adding Gaussian noise at the **input level** (e.g., image), while Align-VL introduces it in the **embedding space** to simulate uncertainty in image-text embeddings, addressing overconfidence from weakly correlated data. Embedding smoothing, rarely mentioned in prior literature, has been occasionally applied in **knowledge graphs**, but it **differs** significantly from its use there, where it enforces semantic smoothness [3]. A similar technique to embedding smoothing is label smoothing. However, unlike label smoothing (unimodal classification), embedding smoothing in Align-VL adjusts the embedding space distribution to alleviate over-reliance on positive and negative pairs, smoothing their similarity distribution and reducing overconfidence, which is crucial for image-text alignment tasks.
> 2. In Align-VL, Random Perturbation and Embedding Smoothing address weakly correlated samples and **overconfidence** by introducing **controlled uncertainty and smoothing in the embedding space, tailored for multimodal alignment**. Unlike traditional **unimodal** methods, these innovations optimize alignment with limited data and resources, enhancing robustness and performance. This unique combination gives Align-VL a significant advantage in resource-constrained scenarios.
>
>
> [3] Guo, S., Wang, Q., Wang, B., Wang, L., & Guo, L. (2015, July). Semantically smooth knowledge graph embedding. In Proceedings of the 53rd Annual Meeting of the Association for Computational Linguistics and the 7th International Joint Conference on Natural Language Processing (Volume 1: Long Papers) (pp. 84-94).

---

> > ### Author Response · Authors · 2024-11-24
> >
> > >W3: The experiment in Table 1 only evaluates a single dataset and compares one baseline, which provides limited evidence.
> >
> > 1. In our experiments, we used **multiple training datasets** including COCO, SBU, and CC3M to validate Align-VL’s performance. We chose the **Flickr dataset** for testing because of its **frequent use in image-text retrieval tasks**, facilitating standardized comparisons with other models. Additionally, we tested on the **MS-COCO dataset to assess Align-VL’s generalization across different scales**, demonstrating its superior performance across various datasets.
> >
> > | Training Dataset Size | Training Dataset | T->I (R@1) |T->I(R@5) | T->I(R@10) | I->T(R@1) | I->T(R@5) | I->T(R@10)|
> > | :-: | :-: | :-: | :-: | :-: |:-: |:-: |:-: |
> > | 560k | COCO (Fusemix)   | 43.68    | 72.53 |82.53  |57.22 |83.4|90.68|
> > | 560k | COCO (Align-VL)   | **45.11**    | **74.16** |**83.05** |**57.76** |**84.38** |**91.6** |
> > | 840k | SUB (Fusemix)  | 22.98    | 47.64 |59.37 |32.36 |56.34 |67.5 |
> > | 840k | SUB (Align-VL)  | **24.35**    | **49** |**60.71** |**31.66** |**56.54** |**67.28** |
> > | 2M | COCO+SBU+VG (Fusemix)   | 43.76   | 72.74|82.36 |60.94 |84.72 |91.68 |
> > | 2M | COCO+SBU+VG (Align-VL)  | **44.27**   | **72.93** |**82.77** |**61.42** |**84.98** |**91.92** |
> >
> > 2. Align-VL aims to optimize **multimodal alignment** while minimizing **data and resource demands**. Evaluating **zero-shot classification** is essential to further validate its generality.
> > - To evaluate Align-VL’s out-of-distribution (OoD) performance on tasks like zero-shot classification and understand its cross-task applicability, we analyzed its performance on **diverse medical datasets** (CheXpert, Shenzhen, IDRiD, Brain Tumor). Specifically, 50 images from different categories were selected as subsets. As shown in the table, Align-VL achieved a **77.21% accuracy on the CheXpert**, significantly outperforming CLIP’s **53.33%** accuracy. On other datasets, Align-VL’s results were **close** to CLIP’s classification performance.
> > - To validate Align-VL’s accuracy on **natural images**, we tested it on **Kaggle’s open-source D&C dataset**. Results showed that CLIP achieved 100.00% accuracy, significantly outperforming **Align-VL’s 67.31%**. These results highlight CLIP’s strength in its advantage of pretraining on a 400M dataset. However, Align-VL demonstrated effectiveness across multiple datasets in zero-shot classification tasks, underscoring its generalization capabilities.
> >
> > | Dataset      | CLIP   | Align-VL |
> > |--------------|--------|----------|
> > | CheXpert     | 53.33  | **77.21**    |
> > | Shenzhen     | 74.47  | 60.21    |
> > | Brain Tumor  | 71.93  | 62.78    |
> > | IDRiD        | 59.18   | 52.33    |
> > | D&C    | 100.00  | 67.31    |
> >
> > With only 3.5M data, Align-VL achieves performance comparable to or even surpassing CLIP on certain datasets, while using **100 times less data** and over 600 times less training time than CLIP, which requires 3000 GPU days and 400M training data. This highlights the importance of Align-VL’s applicability in resource-constrained environments.
> >
> >
> > >W4:The data quality evaluation uses pre-trained CLIP, which may not be ideal given CLIP's own biases, making this evaluation approach potentially limited for assessing data quality.
> >
> > Our evaluation highlights the **insufficient matching** in positive pairs, which is demonstrated through both **qualitative and quantitative** approaches.
> >
> > 1. For the **qualitative** experiments, we conducted **visualization-based assessments** to demonstrate the **low matching degree** in existing image-text paired datasets (As shown in the **table and Figure 4 in main text**, the dataset descriptions differ significantly from human-observed descriptions). In fact, even **humans struggle** to perfectly evaluate the alignment between the two modalities. Therefore, we leveraged the powerful multimodal model **CLIP** to perform the evaluation.
> >
> > | | Sample 1 | Sample 2| Sample 3|
> > | :-: | :-: | :-:|:-: |
> > |Image Caption in Dataset| Green leaves on many trees     | A black metal bench      | Small sailboat in the water     |
> > |  Similarity (CLIP)    | 0.0538     | 0.0699     | 0.0706     |
> > |  Human-observed Description | An orange and blue car    | A person in orange playing tennis     | A flock of sheep  |
> >
> > 2. For the **quantitative** experiments, as the reviewer pointed out, the CLIP model may be influenced by **biases** in its training data when handling image-text pairs. However, its **similarity scores** still partially reflect the degree of matching, as shown in the table. We chose CLIP as a data quality evaluation tool due to its strong performance in multimodal tasks. We did not use the similarity scores computed by CLIP for **further calculations**; they serve **solely as a metric** for quantitatively evaluating the **degree of image-text pairing**.

---

> > > ### Author Response · Authors · 2024-11-26
> > > **Kindly Remind**
> > >
> > > Dear Reviewer rzGx:
> > >
> > > As the review period nears its end, we respectfully await your feedback on our submitted rebuttal. This reminder is shared in hopes of gaining your valuable insights to address and clarify any remaining concerns thoroughly. We sincerely appreciate your efforts in the review process and look forward to your response.
> > >
> > > Best
> > >
> > > The authors

---

### Official Review · Reviewer_eYm7 · 2024-10-30

**Soundness:** 3
**Presentation:** 3
**Contribution:** 3
**Rating:** 6
**Confidence:** 3

**Summary:**

The authors identify issues with current VLMs, such as overconfidence and confusion due to ambiguous samples in datasets with weakly associated, low-quality image-text pairs. To address these challenges, the authors propose Align-VL, a method that enhances multimodal alignment by adjusting the matching degree of data and moderating model overconfidence. Align-VL incorporates two strategies: random perturbation and  embedding smoothing. Align-VL leverages pre-trained unimodal encoders and requires substantially fewer computational resources and data compared to methods like CLIP.  Experiments show that Align-VL outperforms state-of-the-art methods in image-text retrieval tasks and significantly reduces training time and data requirements.

**Strengths:**

1.The descriptions are clear and it is easy to follow.

2. Improvements are achieved on well-known datasets, showing the effectiveness of the proposed method.

**Weaknesses:**

1. The proposed method introduces new parameters, i.e. sigma and alpha. It is needed to select proper parameters.

2. It seems that how performances are effected with varying sigma and alpha.

**Questions:**

See Weaknesses.

---

> ### Author Response · Authors · 2024-11-24
>
> >W1: The proposed method introduces new parameters, i.e. sigma and alpha. It is needed to select proper parameters.
>
> Thank you for your attention to parameter selection! The roles of parameters $\sigma$ and $\alpha$:
> 1. - $\sigma$: Regulates **random perturbation intensity**, introducing noise to simulate uncertainty in embeddings and reduce overconfidence from weakly correlated data. Smaller $\sigma$ ensures **stability**, while larger $\sigma$ risks excessive **uncertainty**.
>
> | Ablation for $\sigma$ | 0.005 | 0.01 | 0.05 | 0.2 |
> |:-: | :-: | :-: | :-:  | :-: |
> | T->I (R@1) | 60.72 | **60.8** |59.08 | 60.06 |
>
>
> - $\alpha$: Controls **embedding smoothing to reduce overconfidence** when handling weakly correlated positive samples. It smooths the embedding space, making similarity predictions for image-text pairs **more robust**. An appropriate $\alpha$ enhances generalization while minimizing reliance on perfectly matched positive samples.
>
> | Ablation for $\alpha$ | 0.01 | 0.05 | 0.1 | 0.2 | 0.3 | Dynamic |
> |:-: | :-: | :-: | :-:  | :-: |:-: |:-: |
> | T->I (R@1) | 58.88 | 59.9 | **60.8** | 60.46 |59.72 | 60.22 |
>
> 2. Parameter Selection: During experiments, we determined the optimal $\sigma$ and $\alpha$ values through **grid search** on the test set to ensure robust alignment. Results showed that $\sigma$ (0.01) and $\alpha$ (0.1) achieved the **best performance** (Dynamic $\alpha$ refers to a strategy where the value of $\alpha$ decreases dynamically as training progresses).
>
>
> >W2: It seems that how performances are effected with varying sigma and alpha.
>
> Thank you for your attention to the impact of parameters $\sigma$ and $\alpha$ on model performance! Below, we provide further clarification on their roles and effects on performance:
>
> 1. Impact of parameter $\sigma$: As the **control parameter** for random perturbation intensity, $\sigma$ influences the model’s robustness in the embedding space. Experimental results indicate:
> - Smaller $\sigma$ values (e.g., 0.005) maintain embedding stability but may be **insufficient** for handling weakly correlated samples.
> - Larger $\sigma$ values (e.g., 0.05 or higher) enhance the model’s adaptability to noise but **may overly disturb the feature space**, negatively impacting alignment accuracy.
>
> | Ablation for sigma | 0.005 | 0.01 | 0.05 | 0.2 |
> |:-: | :-: | :-: | :-:  | :-: |
> | T->I (R@1) | 60.72 | **60.80** |59.08 | 60.06 |
>
> 2. **Impact of parameter $\alpha$**: As the control **parameter for embedding smoothing**, $\alpha$ influences the **distribution smoothness** between positive and negative pairs. An appropriate $\alpha$ value effectively **mitigates overconfidence** issues and enhances generalization:
> - Lower $\alpha$ values (e.g., 0.05) may fail to **adequately reduce** the model’s overconfidence.
> - Higher $\alpha$ values (e.g., 0.3) may make the **model less confident** even for fully matched positive samples, causing **interference with positive pair alignment**.
>
> | Ablation for $\alpha$ | 0.01 | 0.05 | 0.1 | 0.2 | 0.3 | Dynamic |
> |:-: | :-: | :-: | :-:  | :-: |:-: |:-: |
> | T->I (R@1) | 58.88 | 59.9 | **60.8** | 60.46 |59.72 | 60.22 |
>
>
> 3. Parameter search experiments: To quantify the specific impact of $\sigma$ and $\alpha$ on performance, **we conducted experiments to systematically test how different $\sigma$ and $\alpha$ values affect the model’s performance**.
>
> Thank you for the reviewer’s suggestion! We will include the relevant experimental results in the final version to help readers better understand the impact of parameters on model performance.

---

> > ### Author Response · Authors · 2024-11-27
> > **Kindly Remind**
> >
> > Dear Reviewer eYm7,
> >
> > With the discussion period deadline approaching, we kindly request any additional feedback on our rebuttal. Your insights are invaluable, and we appreciate your time and guidance.
> >
> > Best regards,
> >
> > The Authors

---

### Official Review · Reviewer_9rHG · 2024-11-01

**Soundness:** 2
**Presentation:** 2
**Contribution:** 2
**Rating:** 5
**Confidence:** 4

**Summary:**

This paper is based on CLIP and Fusemix and introduces two techniques: Random Perturbation and Embedding Smoothing. The former adds random noise to the output of the unimodal encoder to enhance feature robustness. The latter transforms CLIP's hard labels into soft labels through a target smoothing operation. Experiments on Flickr 30K demonstrate the effectiveness of both techniques. Additionally, the article discusses the issues of partial matches and unmatched pairs in image-text pair data through statistical experiments.

**Strengths:**

1. The methods proposed in this paper are easy to implement, and the experiments can be conducted on a single 3090 GPU, making this paper easy to follow.
2. The experimental results on Flickr30K presented in the paper are promising, showing a significant improvement compared to CLIP trained on the same data (see Table 2).

**Weaknesses:**

1. The experiments presented in the paper are not sufficient and would benefit from the inclusion of additional evaluation datasets and tasks. For instance, incorporating zero-shot image classification results from ImageNet, CUB-200, Oxford Pets, Flowers, and SUN397 could enhance the robustness of the findings.
2. The novelty is limited. Embedding Smoothing closely resembles label smoothing in contrastive learning, while Random Permutation is similar to data augmentation in the feature space.
3. Some figures are not very clear. For example, Fig. 2 should better emphasize the differences between CLIP, FuseMix, and AlignVL.

**Questions:**

1. Are the proposed methods generally effective across more benchmarks, such as the zero-shot image classification datasets mentioned above, as well as additional image-text retrieval datasets?
2. Would you please provide more discussion on the differences between embedding smoothing and label smoothing?

---

> ### Author Response · Authors · 2024-11-24
>
> >W1: The experiments presented in the paper are not sufficient and would benefit from the inclusion of additional evaluation datasets and tasks. For instance, incorporating zero-shot image classification results could enhance the robustness of the findings.
>
> Thank you for the valuable suggestions on the experimental section! Regarding the advice to expand the evaluation datasets and tasks, we have the following responses:
>
> 1.	In our experiments, we used **multiple training datasets** including COCO, SBU, and CC3M to validate Align-VL’s performance. We chose the **Flickr dataset** for testing because of its **frequent use in image-text retrieval tasks**, facilitating standardized comparisons with other models. Additionally, we tested on the **MS-COCO dataset to assess Align-VL’s generalization across different scales**, demonstrating its superior performance across various datasets.
>
> | Training Dataset size | Training Dataset | T->I (R@1) |T->I(R@5) | T->I(R@10) | I->T(R@1) | I->T(R@5) | I->T(R@10)|
> | :-: | :-: | :-: | :-: | :-: |:-: |:-: |:-: |
> | 560k | COCO (Fusemix)   | 43.68    | 72.53 |82.53  |57.22 |83.4|90.68|
> | 560k | COCO (Align-VL)   | **45.11**    | **74.16** |**83.05** |**57.76** |**84.38** |**91.6** |
> | 840k | SUB (Fusemix)  | 22.98    | 47.64 |59.37 |32.36 |56.34 |67.5 |
> | 840k | SUB (Align-VL)  | **24.35**    | **49** |**60.71** |**31.66** |**56.54** |**67.28** |
> | 2M | COCO+SBU+VG (Fusemix)   | 43.76   | 72.74|82.36 |60.94 |84.72 |91.68 |
> | 2M | COCO+SBU+VG (Align-VL)  | **44.27**   | **72.93** |**82.77** |**61.42** |**84.98** |**91.92** |
>
> 2. Align-VL aims to optimize **multimodal alignment** while minimizing **data and resource demands**. Evaluating **zero-shot classification** is essential to further validate its generality.
> - To evaluate Align-VL’s **out-of-distribution** performance on tasks like zero-shot classification and understand its cross-task applicability, we analyzed its performance on **diverse medical datasets** (CheXpert, Shenzhen, IDRiD, Brain Tumor). Specifically, 50 images from different categories were selected as subsets. As shown in the table, Align-VL achieved a **77.21% accuracy on the CheXpert**, significantly outperforming CLIP’s **53.33%** accuracy. On other datasets, Align-VL’s results were **close** to CLIP’s classification performance.
> - To validate Align-VL’s accuracy on **natural images**, we tested it on **Kaggle’s open-source D&C dataset**. Results showed that CLIP achieved 100.00% accuracy, significantly outperforming **Align-VL’s 67.31%**. These results highlight CLIP’s strength in its advantage of pretraining on a 400M dataset. However, Align-VL demonstrated effectiveness across multiple datasets in zero-shot classification tasks, underscoring its generalization capabilities.
>
> | Dataset      | CLIP   | Align-VL |
> |--------------|--------|----------|
> | CheXpert     | 53.33  | **77.21**    |
> | Shenzhen     | 74.47  | 60.21    |
> | Brain Tumor  | 71.93  | 62.78    |
> | IDRiD        | 59.18   | 52.33    |
> | D&C    | **100.00**  | 67.31    |
>
> With only **3.5M data**, Align-VL achieves performance **comparable** to or even surpassing CLIP on certain datasets, while using **100 times less** data and over **600 times less** training time than CLIP, which requires **3000 GPU days and 400M training data**. This highlights the importance of Align-VL’s **applicability in resource-constrained environments**.
>
> Thank you to the reviewer for suggesting additional experiments. We believe these extra tests will further enhance the demonstration of Align-VL’s practicality and robustness.

---

> > ### Author Response · Authors · 2024-11-24
> >
> > >W2: The novelty is limited. Embedding Smoothing closely resembles label smoothing in contrastive learning, while Random Permutation is similar to data augmentation in the feature space.
> >
> > Thank you for your comments! We provide a detailed response regarding the design of Embedding Smoothing and Random Perturbation here:
> >
> > 1.	The **design motivation** and distinction of Embedding Smoothing: Embedding Smoothing, inspired by label smoothing, differs in that it operates within the **embedding space of contrastive learning**, specifically to address **overconfidence** in multimodal alignment tasks. It is designed to smooth the predictive distributions between **positive and negative pairs**, enhancing **adaptability to weakly correlated samples**, which sets it **apart functionally and in application** from traditional label smoothing. This approach is **particularly suited for multimodal tasks**, rather than traditional **unimodal** tasks.
> >
> > 2.	The **uniqueness** of Random Perturbation: Random Perturbation’s uniqueness lies in its primary goal to **simulate uncertainty in data pairs for multimodal alignment tasks**, enhancing model robustness. Unlike typical data augmentation, Align-VL’s Random Perturbation is designed for the embedding space, helping the model learn more “modestly” on weakly correlated datasets. This approach is **particularly suited for multimodal tasks**, rather than traditional unimodal tasks.
> >
> > 3. Innovation summary of Align-VL: The novelty of Align-VL lies in its integration of Embedding Smoothing and Random Perturbation, specifically designed to address challenges like **overconfidence and uncertainty** in multimodal alignment. These strategies advance beyond traditional label smoothing and data augmentation by tailoring to multimodal features, thus enhancing model generalization. Align-VL represents a unique contribution to multimodal alignment, innovating in **both application and implementation**. We thank the reviewer for helping clarify these distinct aspects.
> >
> >
> >
> > >W3: Some figures are not very clear. For example, Fig. 2 should better emphasize the differences between CLIP, FuseMix, and AlignVL.
> >
> > Thank you for the feedback on the clarity of our illustrations! Regarding the display of differences between methods in Figure 2, we will make the following improvements:
> >
> > 1. CLIP for **Contrastive Learning**, FuseMix for Enhanced Mixing Embeddings, and Align-VL for **Moderating Model Overconfidence and Enhancing Robustness**. In Fusemix and Align-VL, the text anchor and visual positive sample are derived from **mixed features**. In Align- VL, the visual and text positive pairs are the embedding augmented with **random perturbation**.
> >
> > 2. Enhance visual contrast: We will adjust the **visual representation** of CLIP, FuseMix, and Align-VL in Figure 2 by using unique colors or borders for each method to emphasize their differences. **Clear labels** will also be added to help readers easily identify the distinctive features of each approach.

---

> ### Author Response · Authors · 2024-11-24
>
> >Q1: Are the proposed methods generally effective across more benchmarks, such as the zero-shot image classification datasets mentioned above, as well as additional image-text retrieval datasets?
>
>
> Thank you for highlighting the broad applicability of our method! Regarding Align-VL’s performance on additional benchmarks, we have the following responses and plans:
>
> 1.	In our experiments, we used **multiple training datasets** including COCO, SBU, and CC3M to validate Align-VL’s performance. We chose the **Flickr dataset** for testing because of its **frequent use in image-text retrieval tasks**, facilitating standardized comparisons with other models. Additionally, we tested on the **MS-COCO dataset (25K) to assess Align-VL’s generalization across different scales**, demonstrating its superior performance across various datasets.
>
> | Training Dataset Size | Training Dataset | T->I (R@1) |T->I(R@5) | T->I(R@10) | I->T(R@1) | I->T(R@5) | I->T(R@10)|
> | :-: | :-: | :-: | :-: | :-: |:-: |:-: |:-: |
> | 560k | COCO (Fusemix)   | 43.68    | 72.53 |82.53  |57.22 |83.4|90.68|
> | 560k | COCO (Align-VL)   | **45.11**    | **74.16** |**83.05** |**57.76** |**84.38** |**91.6** |
> | 840k | SUB (Fusemix)  | 22.98    | 47.64 |59.37 |32.36 |56.34 |67.5 |
> | 840k | SUB (Align-VL)  | **24.35**    | **49** |**60.71** |**31.66** |**56.54** |**67.28** |
> | 2M | COCO+SBU+VG (Fusemix)   | 43.76   | 72.74|82.36 |60.94 |84.72 |91.68 |
> | 2M | COCO+SBU+VG (Align-VL)  | **44.27**   | **72.93** |**82.77** |**61.42** |**84.98** |**91.92** |
>
> 2. Align-VL aims to optimize **multimodal alignment** while minimizing **data and resource demands**. Evaluating **zero-shot classification** is essential to further validate its generality.
> - To evaluate Align-VL’s **out-of-distribution** performance on tasks like zero-shot classification and understand its cross-task applicability, we analyzed its performance on **diverse medical datasets** (CheXpert, Shenzhen, IDRiD, Brain Tumor). Specifically, 50 images from different categories were selected as subsets. As shown in the table, Align-VL achieved a **77.21% accuracy on the CheXpert**, significantly outperforming CLIP’s **53.33%** accuracy. On other datasets, Align-VL’s results were **close** to CLIP’s classification performance.
> - To validate Align-VL’s accuracy on **natural images**, we tested it on **Kaggle’s open-source D&C dataset**. Results showed that CLIP achieved 100.00% accuracy, significantly outperforming **Align-VL’s 67.31%**. These results highlight CLIP’s strength in its advantage of pretraining on a 400M dataset. However, Align-VL demonstrated effectiveness across multiple datasets in zero-shot classification tasks, underscoring its generalization capabilities.
>
> | Dataset      | CLIP   | Align-VL |
> |--------------|--------|----------|
> | CheXpert     | 53.33  | **77.21**    |
> | Shenzhen     | 74.47  | 60.21    |
> | Brain Tumor  | 71.93  | 62.78    |
> | IDRiD        | 59.18   | 52.33    |
> | D&C    | 100.00  | 67.31    |
>
> With only **3.5M data**, Align-VL achieves performance **comparable** to or even surpassing CLIP on certain datasets, while using **100 times less** data and over **600 times less** training time than CLIP, which requires **3000 GPU days and 400M training data**. This highlights the importance of Align-VL’s **applicability in resource-constrained environments**.

---

> > ### Author Response · Authors · 2024-11-24
> >
> > >Q2: Would you please provide more discussion on the differences between embedding smoothing and label smoothing?
> >
> >
> > Thank you for the reviewer’s attention to the differences between Embedding Smoothing and Label Smoothing! We further discuss the distinctions in their design purposes and applicable scenarios:
> > 1. **Different design purposes**: Label Smoothing is designed to **reduce model overconfidence** in specific categories in **classification** tasks by allocating slightly less than 100% probability to the target category and spreading the remainder across other categories. In contrast, Embedding Smoothing aims to **smooth the embedding space in multimodal alignment tasks**, addressing overconfidence with **weakly correlated positive samples**. It enhances **robustness** not by adjusting labels but by smoothing the entire embedding space, improving the model’s predictions of similarity between image and text pairs.
> > 2. **Different application scenarios**: Label Smoothing is typically applied in **unimodal classification** tasks to optimize model decision-making by adjusting the distribution of category labels. Embedding Smoothing, however, is more suited to **multimodal alignment tasks**. It operates on the **similarity distribution** between image and text embeddings, reducing the model’s **overreliance on positive pairs** when dealing with weakly correlated samples and thus **enhancing generalization** to unseen samples.
> > 3. **Application differences in contrastive learning**: Embedding Smoothing operates across the **entire embedding space**, beyond just label probability adjustments. It assigns a minor non-zero probability to all negative samples, promoting a smoother embedding distribution during similarity calculations, thus **significantly reducing overconfidence** in multimodal tasks. Conversely, Label Smoothing is less effective in contrastive learning contexts.
> >
> > In summary, although Embedding Smoothing and Label Smoothing are similarly motivated, they **differ significantly in their application scenarios, targets, and mechanisms**. We will clarify these distinctions further in our paper and appreciate the reviewer’s prompt to discuss this key aspect in more detail.

---

> > > ### Author Response · Authors · 2024-11-27
> > > **Kindly Remind**
> > >
> > > Dear Reviewer 9rHG,
> > >
> > > With the ICLR rebuttal deadline approaching, we are eager to hear your feedback and finalize our submission with your valued insights. Your expertise is crucial at this pivotal moment, and we greatly appreciate your time and support.
> > >
> > > Thank you for your thoughtful consideration.
> > >
> > > Warm regards,
> > > Authors

---

### Official Review · Reviewer_9rVi · 2024-11-02

**Soundness:** 2
**Presentation:** 3
**Contribution:** 2
**Rating:** 5
**Confidence:** 4

**Summary:**

The paper presents Align-VL, a novel approach to enhance multimodal alignment in Vision-Language Models (VLMs) by addressing the issues of overconfidence and ambiguous training data pairs. Current VLMs like CLIP encounter performance degradation when trained on weakly associated, low-quality image-text pairs, often becoming overconfident and confused by ambiguous samples. Align-VL introduces two main techniques to tackle these challenges: Random Perturbation and Embedding Smoothing. Random Perturbation introduces noise to both visual and textual embeddings during training to enhance generalization, while Embedding Smoothing reduces model overconfidence by ensuring smoother and less peaked prediction distributions. The paper demonstrates that Align-VL outperforms state-of-the-art (SoTA) methods in image-text retrieval tasks, while significantly reducing data requirements and training time. The proposed method aims to be more computationally efficient by leveraging pre-trained unimodal encoders and improving the quality of multimodal alignment.

**Strengths:**

- The paper proposes a creative enhancement for vision-language alignment through the use of Random Perturbation and Embedding Smoothing applied to the latent space of pre-trained unimodal encoders. While these techniques were introduced from existing methods in other fields, their combination and application in this context is effective in moderating overconfidence, adding an extra layer of robustness to multimodal systems.
- The paper is clearly written, with well-structured sections that logically present the motivation, proposed methods, and experimental results. The equations and theoretical analysis are well-defined and helpful for understanding the concepts. However, some figures are confusing and could be better explained.

**Weaknesses:**

- A key limitation is the lack of evaluation on larger-scale datasets comparable to those used by models like CLIP (e.g., 400 million data pairs). The paper acknowledges this but does not include any further experiments or discussions on large-scale datasets to substantiate the scalability of Align-VL. There are publicly available large-scale text-image datasets, such as the LAION dataset, which could have been used to provide additional insights. Including a discussion on potential challenges or a small-scale simulated analysis could strengthen this point.
- The model evaluation is somewhat limited in scope. First, the authors only used the Flickr dataset for evaluation, which is insufficient to assess the model's general performance across a broader range of data. It would be better to include additional datasets, such as MSCOCO, to provide a more comprehensive evaluation. Second, the evaluation was restricted to the image and text retrieval task. Evaluating the model's performance on other tasks, such as zero-shot classification, would provide a more complete understanding of its capabilities.

**Questions:**

- The method has not been tested on extremely large datasets. Could the authors elaborate on the potential challenges of scaling Align-VL to datasets with large scale datasets? Also,
- Random Perturbation appears similar to other types of regularization techniques, such as dropout or mixup. How does Align-VL compare to these standard methods? Would substituting Random Perturbation with another noise-based regularization method yield similar results?
- I would like to hear the authors' opinion on whether the proposed method would be effective for training a multimodal model from scratch rather than leveraging pre-trained unimodal encoders.
- Figures are somewhat confusing.
  - In Figure 2, It is confusing what the grey square box, "Negative and "Positive mean. Also, what do blurred embeddings and small grey rectangles mean?
  - In Figure 3, it looks like random perturbation is used only for text embedding and the ES is used only for image embedding. Also, those embeddings from A_x and A_y are fed into contrastive loss without ES. Please clarify it.
- How does the performance differ by the batch size of N because the smoothed target distribution in the Embedding Smoothing is defined with N? It would be great to see an ablation study by N.

**Details Of Ethics Concerns:**

There do not appear to be any significant ethical concerns with this work.

---

> ### Author Response · Authors · 2024-11-24
>
> >W1: A key limitation is the lack of evaluation on larger-scale datasets comparable to those used by models like CLIP (e.g., 400 million data pairs). The paper acknowledges this but does not include any further experiments or discussions on large-scale datasets to substantiate the scalability of Align-VL. There are publicly available large-scale text-image datasets, such as the LAION dataset, which could have been used to provide additional insights. Including a discussion on potential challenges or a small-scale simulated analysis could strengthen this point.
>
> Thank you for the reviewer’s suggestions! While we may not have sufficient time to complete experiments on large-scale datasets, we conducted a **small-scale analysis** on the **LAION-2M** (including COCO), increasing data pairs to assess performance and training time. Results showed a significant increase in training days and demonstrated that Align-VL consistently **exceeds the baseline Fusemix** across various data volumes under **resource constraints**.
>
> | Training Dataset | Fusemix | Align-VL |Training Days |
> | :-: | :-: | :-: |:-: |
> | LAION-2M     | 61.40 | **62.10**  | 4   |
> | COCO     | 57.80     | **60.80**  |1.4  |
> | SBU     | 43.32     | **47.80**  |1.7  |
> | VG     | 52.90     | **51.66**  | 1.8  |
>
> Our proposed Align-VL aims to enhance **computational and data efficiency** for modal alignment in **resource-efficient** settings. It has outperformed state-of-the-art methods on public datasets, notably improving the R@1 score in retrieval tasks. With only 3.5M data, Align-VL exceeds CLIP’s performance on the **MS COCO dataset** and does so with **100 times less data and over 600 times less training time than CLIP**, which requires **3000 GPU days and 400M training data**.
>
> |     | Text->Image | Image->Text | Training Dataset Size |GPU Days |
> | :-: | :-: | :-:  |:-:  |:-:  |
> | CLIP  | 37.8%  | 58.4%  | 400M  | 3000 |
> | ALIGN | 45.6% | 48.6% | 1B  | \ |
> | LIT   | 43.6 | 59.5%  | 4B  | \ |
> | Align-VL| **45.86%** | **62.86%**  | **3.5M**  |**5**  |
>
> We were unable to complete experiments on large-scale datasets for two reasons. However, we will include the results from the large-scale datasets in our submission as soon as they are completed.
>
> 1. Powerful models require extensive datasets and significant **computational resources**, making them costly for practical applications. Recent successes in multimodal alignment rely on **intensive GPU use and large datasets**, such as CLIP’s 3000 GPU Days, which are **impractical in resource-limited settings**. Thus, creating **cost-effective frameworks** for multimodal alignment is crucial.
>
> 2. Align-VL faces challenges in scalability on larger datasets, including **1) increased computational demands** (potentially thousands of GPU days for datasets with hundreds of millions of data points), **2) scarcity of high-quality multimodal pairs**, and large proportions of noisy data. However, its design reduces computational needs and improves **robustness** to such data challenges.

---

> > ### Author Response · Authors · 2024-11-24
> >
> > >W2: The model evaluation is somewhat limited in scope. First, the authors only used the Flickr dataset for evaluation, which is insufficient to assess the model's general performance across a broader range of data. It would be better to include additional datasets, such as MSCOCO, to provide a more comprehensive evaluation. Second, the evaluation was restricted to the image and text retrieval task. Evaluating the model's performance on other tasks, such as zero-shot classification, would provide a more complete understanding of its capabilities.
> >
> > 1.	In our experiments, we used **multiple training datasets** including COCO, SBU, and CC3M to validate Align-VL’s performance. We chose the **Flickr dataset** for testing because of its **frequent use in image-text retrieval tasks**, facilitating standardized comparisons with other models. Additionally, we tested on the **MS-COCO dataset to assess Align-VL’s generalization across different scales**, demonstrating its superior performance across various datasets.
> >
> > | Training Dataset Size | Training Dataset | T->I (R@1) |T->I(R@5) | T->I(R@10) | I->T(R@1) | I->T(R@5) | I->T(R@10)|
> > | :-: | :-: | :-: | :-: | :-: |:-: |:-: |:-: |
> > | 560k | COCO (Fusemix)   | 43.68    | 72.53 |82.53  |57.22 |83.4|90.68|
> > | 560k | COCO (Align-VL)   | **45.11**    | **74.16** |**83.05** |**57.76** |**84.38** |**91.6** |
> > | 840k | SUB (Fusemix)  | 22.98    | 47.64 |59.37 |32.36 |56.34 |67.5 |
> > | 840k | SUB (Align-VL)  | **24.35**    | **49** |**60.71** |**31.66** |**56.54** |**67.28** |
> > | 2M | COCO+SBU+VG (Fusemix)   | 43.76   | 72.74|82.36 |60.94 |84.72 |91.68 |
> > | 2M | COCO+SBU+VG (Align-VL)  | **44.27**   | **72.93** |**82.77** |**61.42** |**84.98** |**91.92** |
> >
> >
> > 2. Align-VL aims to optimize **multimodal alignment** while minimizing **data and resource demands**. Evaluating **zero-shot classification** is essential to further validate its generality.
> > - To evaluate Align-VL’s **out-of-distribution** (OoD) performance on tasks like zero-shot classification and understand its cross-task applicability, we analyzed its performance on **diverse medical datasets** (CheXpert, Shenzhen, IDRiD, Brain Tumor). Specifically, 50 images from different categories were selected as subsets. As shown in the table, Align-VL achieved a **77.21% accuracy on the CheXpert**, significantly outperforming CLIP’s **53.33%** accuracy. On other datasets, Align-VL’s results were **close** to CLIP’s classification performance.
> > - To validate Align-VL’s accuracy on **natural images**, we tested it on **Kaggle’s open-source D&C dataset**. Results showed that CLIP achieved 100.00% accuracy, significantly outperforming **Align-VL’s 67.31%**. These results highlight CLIP’s strength in its advantage of pretraining on a 400M dataset. However, Align-VL demonstrated effectiveness across multiple datasets in zero-shot classification tasks, underscoring its generalization capabilities.
> >
> > | Dataset      | CLIP   | ALIGN-VL |
> > |--------------|--------|----------|
> > | CheXpert     | 53.33  | **77.21**    |
> > | Shenzhen     | 74.47  | 60.21    |
> > | Brain Tumor  | 71.93  | 62.78    |
> > | IDRiD        | 59.18   | 52.33    |
> > | D&C    | **100.00**  | 67.31    |
> >
> > With only **3.5M data**, Align-VL achieves performance **comparable** to or even surpassing CLIP on certain datasets, while using **100 times less** data and over **600 times less** training time than CLIP, which requires **3000 GPU days and 400M training data**. This highlights the importance of Align-VL’s **applicability in resource-constrained environments**.

---

> ### Author Response · Authors · 2024-11-24
>
> >Q1: The method has not been tested on extremely large datasets. Could the authors address potential challenges in scaling Align-VL to such datasets? Additionally, Random Perturbation resembles regularization techniques like dropout or mixup. How does Align-VL compare to these methods, and would substituting Random Perturbation with another regularization approach yield similar results?
>
> 1. Align-VL faces challenges in scalability on larger datasets, including
> - Despite using **lightweight V-L Adapters** and frozen unimodal encoders to reduce computational demands, Align-VL still faces significant increases in **computing and storage needs** when processing datasets with hundreds of millions of data points, potentially requiring **thousands of GPU days**. Furthermore, its “embedding smoothing” technique, which adjusts the probability distribution for each batch, **incurs overhead on large-scale datasets**.
> - On very large datasets, the variability in data quality leads to increased **noise and mismatched samples**, amplifying model overconfidence. This requires fine-tuning **“embedding smoothing” and “random perturbation” parameters** to enhance the model’s **generalization and stability**.
>
> However, the current design of Align-VL also **reduces computational needs and improves robustness** to such data challenges. Additionally, it is worth noting that in tests conducted on the 25K **MS-COCO** retrieval dataset, Align-VL **surpassed the retrieval performance** of much larger datasets, as illustrated in the table.
>
> |     | Text->Image | Image->Text | Training Dataset Size |GPU Days |
> | :-: | :-: | :-:  |:-:  |:-:  |
> | CLIP  | 37.8%  | 58.4%  | 400M  | 3000 |
> | ALIGN | 45.6% | 48.6% | 1B  | \  |
> | LIT   | 43.6 | 59.5%  | 4B  | \  |
> | Align-VL| **45.86%** | **62.86%**  | **3.5M**  |**5**  |
>
>
> 2. Comparison of random perturbation with **mixup regularization** techniques:
> 	• In Align-VL, “random perturbation” is designed not merely for adding noise but to **simulate uncertainty** in image-text embedding spaces, addressing **overconfidence from weakly correlated data**. This is **distinct from dropout or mixup**, which aim primarily to prevent model overfitting.
> 	• Compared to dropout and mixup: Dropout is **already integrated into both baseline and Align-VL training**. Although mixup was tested, it proved less effective than **“random perturbation” for multimodal alignment tasks (In Table, Text->Image R@1)**. Align-VL’s perturbation strategy specifically targets minor mismatches between images and texts, unlike mixup, which increases **data diversity** but does **not reduce model overconfidence**.
>
>
> | Training Dataset | Mixup | Align-VL |
> | :-: | :-: | :-: |
> | COCO     | 57.80  | **60.80** |
> | SBU     | 43.32   | **47.80**    |
> | VG     | 51.66    | **52.90**    |
>
>
> >Q2: I would like to hear the authors' opinion on whether the proposed method would be effective for training a multimodal model from scratch rather than leveraging pre-trained unimodal encoders.
>
>
> Thank you for the **valuable questions** raised by the reviewer! Align-VL is designed to use pre-trained unimodal encoders for efficient multimodal alignment. For training from scratch, adjustments and optimizations may be necessary. Regarding the issue of training multimodal models from scratch, we have the following perspectives:
>
> 1.	The core design of Align-VL involves adding **lightweight adapters** to pre-trained unimodal encoders, enabling efficient multimodal alignment with reduced computational and data needs. This approach leverages existing unimodal models to **minimize the costs of training from scratch**, making Align-VL ideal for **resource-limited settings** while still delivering robust performance.
>
> 2.	Challenges of training from scratch: If Align-VL were trained entirely from scratch, its design might need adjustments. It relies on **strong unimodal representations from pre-trained encoders**, which **ground-up models might initially lack**, especially for complex image-text alignments. Thus, Align-VL’s “random perturbation” and “embedding smoothing” strategies may not perform as effectively in early training stages.
>
> 3.	Adaptability and exploration: While Align-VL currently utilizes pre-trained unimodal encoders, its core strategies—optimizing alignment and reducing overconfidence via perturbations and embedding smoothing—are **potentially adaptable** to training from scratch. Our preliminary results, as detailed in the table, suggest that Align-VL **remains effective even when built from the ground up**.
>
> | Training Dataset (COCO) | T->I(R@1) | I->T(R@1) |
> | -------- | -------- | -------- |
> | Align-VL (training from scratch) w/o RP and ES  | 54.2     | 65.8  |
> | Align-VL (training from scratch)  | **54.4**     | **66.4**  |
> | Align-VL w/o RP and ES  | 57.80    | 71.60  |
> | Align-VL  | **65.72**     | **81.60**  |

---

> > ### Author Response · Authors · 2024-11-24
> >
> > >Q3: Figures are somewhat confusing. In Figure 2, It is confusing what the grey square box, "Negative and "Positive mean. Also, what do blurred embeddings and small grey rectangles mean?
> >
> > In Figure 2, the triangular boxes labeled **“Positive” and “Negative”** represent the types of **sample pairs in multimodal alignment tasks—matching and mismatched pairs**, respectively. The grey box indicates **perturbation** that for positive pairs, both visual and textual features are perturbed to simulate **uncertainty** in image-text embedding spaces.
> >
> > Blurred embeddings and small grey rectangles in Align-VL represent **noise from “random perturbation”** to enhance generalization. This noise aids robustness against weakly correlated or uncertain samples. The small grey rectangles show **noise in the perturbed embedding space**, reducing **overconfidence** in embeddings for matched pairs (positive samples).
> >
> > We will further optimize the illustrations to more clearly convey the key concepts of our method as you suggested.
> >
> >
> > >Q4: In Figure 3, it looks like random perturbation is used only for text embedding and the ES is used only for image embedding. Also, those embeddings from A_x and A_y are fed into contrastive loss without ES. Please clarify it.
> >
> > 1. In Figure 3, Random Perturbation (RP) and Embedding Smoothing (ES) are applied **simultaneously to both image and text embeddings** in Align-VL. This integration enhances adaptability to weakly **correlated samples** by introducing perturbations into both embedding spaces. Embedding smoothing is also implemented across modalities to smooth **contrastive loss** between positive and negative pairs, **reducing model overconfidence**.
> > 2.	In the diagram, embeddings from A_x and A_y are subjected to Random Perturbation and Embedding Smoothing **before entering the contrastive loss function**. This smoothing adjusts the **final embedding distribution**, ensuring the model maintains balanced biases across samples in the contrastive learning process, **consistently applied** to both text and image embeddings.
> >
> > Thank you for identifying this **potential visual misunderstanding**. We will optimize the representation in Figure 3 to prevent any confusion in our revised submission.
> >
> >
> > >Q5: How does the performance differ by the batch size of N because the smoothed target distribution in the Embedding Smoothing is defined with N? It would be great to see an ablation study by N.
> >
> > To systematically investigate the impact of **batch size on performance**, we plan to conduct ablation experiments with **varying batch sizes (1k, 2k, 5k, 10k, and 15k)**. We will evaluate how these changes affect Align-VL’s performance, especially on text-to-image and image retrieval metrics, to better understand the role of batch size in embedding smoothing. Our results indicate that a **batch size of 10k** offers optimal performance.
> >
> > | Ablation for N | 1k | 2k | 5k | 10k | 15k |
> > |:-: | :-: | :-: | :-:  | :-: |:-: |
> > | T->I (R@1) | 58.3 | 59.08 |59.3 | **60.8** |59.38 |
> > | I->T (R@1) | 70 | 71.7 |**73.1** | 72.6 |73 |
> > | Average | 64.15 | 65.39 | 66.2 | **66.7** | 66.19 |
> >
> > We will include the results of these ablation experiments in our paper and discuss the impact of batch size on model performance.

---

> > > ### Author Response · Authors · 2024-11-26
> > > **Kindly Remind**
> > >
> > > Dear Reviewer 9rVi,
> > >
> > > This is a gentle reminder that the deadline for the discussion period is approaching fast. We would greatly appreciate any additional feedback you could provide on our rebuttal, recognizing the tight schedules you manage. Your guidance is invaluable to enhancing our submission. We look forward to your expert insights.
> > >
> > > Best
> > >
> > > The authors

---

### Author Response · Authors · 2024-11-30

Dear Reviewers,

Thank you for your valuable insights on our submission. We have carefully addressed your comments in our rebuttal and eagerly await your feedback. As the ICLR decision timeline is approaching, could you please review our responses at your earliest convenience? Your prompt feedback would be invaluable to us.

Best regards,

The authors

---

### Author Response · Authors · 2024-12-01

We really appreciate all reviewers for their valuable feedback. Our code will be released soon.

We greatly appreciate the reviewers’ recognition of our contributions (9rVi, 9rHG, eYm7, rzGx). Notably, Align-VL is highlighted for its **effectiveness in moderating overconfidence** (9rVi) and for being clearly written (9rVi, rzGx). The method is **efficient, capable of running on a single 3090 GPU** (9rHG), while demonstrating compelling **performance improvements** (9rHG, eYm7). Additionally, Align-VL provides a **valuable insight** into the reliance of current contrastive learning methods on single positive pairs (rzGx).

In our individual replies, we attempted to address specific questions and comments as clearly and detailed as possible. Here, we briefly summarize these additional experiments and evaluations:

- We evaluated Align-VL on the **MS-COCO dataset**, demonstrating it exceeds CLIP’s performance while using **100 times** less data and over **600 times less** training time.

- To evaluate Align-VL’s **zero-shot** classification performance, we tested it on **four medical datasets** and one **natural** image dataset, where its results were slightly behind CLIP’s classification performance.

- We trained Align-VL’s unimodal encoder **from scratch**, showing reduced gains but confirming its effectiveness even when built from the ground up.

- We conducted a small-scale analysis on **LAION-2M**, revealing that Align-VL consistently outperforms Fusemix across various data volumes under resource constraints.

- We conducted an **ablation** study on **$N$, $\sigma$, and $\alpha$**, thoroughly analyzing their impact on Align-VL’s performance.

We hope these additional results further establish Align-VL as the state-of-the-art in efficient alignment for Vision-Language Models.

---

### Author Response · Authors · 2024-12-02

Dear Reviewers,

Serving as a reviewer is vital to advancing deep learning research and a key responsibility for researchers. We understand your busy schedule, but we sincerely hope you can take a moment to review the rebuttal we submitted to address your concerns. We kindly request your thorough consideration.

The authors

---

### Meta-Review · Area_Chair_a47j · 2024-12-20

**Metareview:**

The reviewers acknowledge the benefits of this work with a simple idea and good presentation quality. However, reviewers raised several concerns regarding limited experiments and evaluation scope, unclear motivation, and lack of technical novelty. Although the authors attempted to address these issues, the scope of the experiments is still too small in scale and offers limited comparisons with baselines, making it difficult to assess the method's generalizability and scalability. While the reviewers did not engage during the rebuttal period, the AC concludes that the raised issues were not adequately addressed and, therefore, recommends rejection.

**Additional Comments On Reviewer Discussion:**

- Reviewer 9rVi, Reviewer 9rHG, and Reviewer rzGx raised concerns about the limited scope of the experiments and evaluations of the paper.
- Reviewer 9rHG and Reviewer rzGx are concerned about the limited technical novelty of the paper.
- Reviewer eYm only requested the analysis on the hyperparameters.
- Despite the authors' efforts in their rebuttal, unfortunately, all the reviewers have not engaged in the discussion
- After reading the rebuttal, the AC concluded that the paper still needs significant improvement to be published. "

---

### Decision · Program_Chairs · 2025-01-22

Reject